# Sinkhorn Barycenter via Functional Gradient Descent

**Zebang Shen**[*]    **Zhenfu Wang**[†]    **Alejandro Ribeiro**[*]    **Hamed Hassani**[*]
[*]Department of Electrical and Systems Engineering    [†]Department of Mathematics
University of Pennsylvania
{zebang@seas,zwang423@math,aribeiro@seas,hassani@seas}.upenn.edu

## Abstract

In this paper, we consider the problem of computing the barycenter of a set of probability distributions under the Sinkhorn divergence. This problem has recently found applications across various domains, including graphics, learning, and vision, as it provides a meaningful mechanism to aggregate knowledge. Unlike previous approaches which directly operate in the space of probability measures, we recast the Sinkhorn barycenter problem as an instance of unconstrained functional optimization and develop a novel functional gradient descent method named `Sinkhorn Descent` (SD). We prove that SD converges to a stationary point at a sublinear rate, and under reasonable assumptions, we further show that it asymptotically finds a global minimizer of the Sinkhorn barycenter problem. Moreover, by providing a mean-field analysis, we show that SD preserves the weak convergence of empirical measures. Importantly, the computational complexity of SD scales linearly in the dimension $d$ and we demonstrate its scalability by solving a 100-dimensional Sinkhorn barycenter problem.

## 1   Introduction

Computing a nonlinear interpolation between a set of probability measures is a foundational task across many disciplines. This problem is typically referred as the barycenter problem and, as it provides a meaningful metric to aggregate knowledge, it has found numerous applications. Examples include distribution clustering [Ye et al., 2017], Bayesian inference [Srivastava et al., 2015], texture mixing [Rabin et al., 2011], and graphics [Solomon et al., 2015], etc. The barycenter problem can be naturally cast as minimization of the average distance between the target measure (barycenter) and the source measures; and the choice of the distance metric can significantly impact the quality of the barycenter [Feydy et al., 2019]. In this regard, the Optimal Transport (OT) distance (a.k.a. the Wasserstein distance) and its entropy regularized variant (a.k.a. the Sinkhorn divergence) are the most suitable geometrically-faithful metrics, while the latter is more computational friendly. In this paper, we provide efficient and provable methods for the Sinkhorn barycenter problem.

The prior work in this domain has mainly focused on finding the barycenter by optimizing directly in the space of (discrete) probability measures. We can divide these previous methods into three broad classes depending on how the support of the barycenter is determined:
(i) The first class assumes a fixed and prespecified support set for the barycenter and only optimizes the corresponding weights [Staib et al., 2017, Dvurechenskii et al., 2018, Kroshnin et al., 2019]. Accordingly, the problem reduces to minimizing a convex objective subject to a simplex constraint. However, fixing the support without any prior knowledge creates undesired bias and affects the quality of the final solution. While increasing the support size (possibly exponentially in the dimension $d$) can help to mitigate the bias, it renders the procedure computationally prohibitive as $d$ grows.
(ii) To reduce the bias, the second class considers optimizing the support and the weights through an alternating procedure [Cuturi and Doucet, 2014, Claici et al., 2018]. Since the barycenter objective is

not jointly convex with respect to the support and the weights, these methods in general only converge to a stationary point, which can be far from the true minimizers.

(iii) Unlike the aforementioned classes, Luise et al. [2019] recently proposed a conditional gradient method with a growing support set. This method enjoys sublinear convergence to the global optimum under the premise that a $d$-dimensional nonconvex subproblem can be globally minimized per-iteration. However, nonconvex optimization is generally intractable in high dimensional problems (large $d$) and only stationary points can be efficiently reached. Hence, the guarantee of [Luise et al., 2019] has limited applicability as the dimension grows.

In this paper, we provide a new perspective on the Sinkhorn barycenter problem: Instead of operating in the space of probability measures, we view the barycenter as the push-forward measure of a given initial measure under an unknown mapping. We thus recast the barycenter problem as an unconstrained functional optimization over the space of mappings. Equipped with this perspective, we make the following contributions:

- We develop a novel functional gradient descent method, called `Sinkhorn Descent` (`SD`), which operates by finding the push-forward mapping in a Reproducing Kernel Hilbert Space that allows the fastest descent, and consequently solves the Sinkhorn barycenter problem iteratively. We then define the Kernelized Sinkhorn Barycenter Discrepancy (KSBD) to characterize the non-asymptotic convergence of `SD`. In particular, we prove that KSBD vanishes under the `SD` iterates at the rate of $\mathcal{O}(\frac{1}{t})$, where $t$ is the iteration number.

- We prove that `SD` preserves the weak convergence of empirical measures. Concretely, use $\mathtt{SD}^t(\cdot)$ to denote the output of `SD` after $t$ iterations and let $\alpha_N$ be an empirical measure of $\alpha$ with $N$ samples. We have $\lim_{N\to\infty} \mathtt{SD}^t(\alpha_N) = \mathtt{SD}^t(\alpha)$. Such asymptotic analysis allows us to jointly study the behavior of `SD` under either discrete or continuous initialization.

- Under a mild assumption, we prove that KSBD is a valid discrepancy to characterize the optimality of the solution, i.e. the vanishing of KSBD implies the output measure of `SD` converges to the global optimal solution set of the Sinkhorn barycenter problem.

Further, we show the efficiency and efficacy of `SD` by comparing it with prior art on several problems. We note that the computation complexity of `SD` depends *linearly* on the dimension $d$. We hence validate the scalability of `SD` by solving a 100-dimensional barycenter problem, which cannot be handled by previous methods due to their exponential dependence on the problem dimension.

**Notations.** Let $\mathcal{X} \subseteq \mathbb{R}^d$ be a compact ground set, endowed with a symmetric ground metric $c : \mathcal{X} \times \mathcal{X} \to \mathbb{R}_+$. Without loss of generality, we assume $c(x, y) = \infty$ if $x \notin \mathcal{X}$ or $y \notin \mathcal{X}$. We use $\nabla_1 c(\cdot, \cdot) : \mathcal{X}^2 \to \mathcal{X}$ to denote its gradient w.r.t. its first argument. Let $\mathcal{M}_1^+(\mathcal{X})$ and $\mathcal{C}(\mathcal{X})$ be the space of probability measures and continuous functions on $\mathcal{X}$. We denote the support for a probability measure $\alpha \in \mathcal{M}_1^+(\mathcal{X})$ by $\mathrm{supp}(\alpha)$ and we use $\alpha - a.e.$ to denote "almost everywhere w.r.t. $\alpha$". For a vector $\mathbf{a} \in \mathbb{R}^d$, we denote its $\ell_2$ norm by $\|\mathbf{a}\|$. For a function $f : \mathcal{X} \to \mathbb{R}$, we denote its $L^\infty$ norm by $\|f\|_\infty := \max_{x \in \mathcal{X}} |f(x)|$ and denote its gradient by $\nabla f(\cdot) : \mathcal{X} \to \mathbb{R}^d$. For a vector function $f : \mathcal{X} \to \mathbb{R}^d$, we denote its $(2, \infty)$ norm by $\|f\|_{2,\infty} := \max_{x \in \mathcal{X}} \|f(x)\|$. For an integer $n$, denote $[n] := \{1, \cdots, n\}$.

Given an Reproducing Kernel Hilbert Space (RKHS) $\mathcal{H}$ with a kernel function $k : \mathcal{X} \times \mathcal{X} \to \mathbb{R}_+$, we say a vector function $\psi = [[\psi]_1, \cdots, [\psi]_d] \in \mathcal{H}^d$ if each component $[\psi]_i$ is in $\mathcal{H}$. The space $\mathcal{H}$ has a natural inner product structure and an induced norm, and so does $\mathcal{H}^d$, i.e. $\langle f, g \rangle_{\mathcal{H}^d} = \sum_{i=1}^d \langle [f]_i, [g]_i \rangle_{\mathcal{H}}, \forall f, g \in \mathcal{H}^d$ and the norm $\|f\|_{\mathcal{H}^d}^2 = \langle f, f \rangle_{\mathcal{H}^d}$. The reproducing property of the RKHS $\mathcal{H}$ reads that given $f \in \mathcal{H}^d$, one has $[f]_i(x) = \langle [f]_i, k_x \rangle_{\mathcal{H}}$ with $k_x(y) = k(x, y)$, which by Cauchy-Schwarz inequality implies that there exists some constant $M_{\mathcal{H}} > 0$ such that

$$\|f\|_{2,\infty} \le M_{\mathcal{H}} \|f\|_{\mathcal{H}^d}, \forall f \in \mathcal{H}^d. \tag{1}$$

Additionally, for a functional $F : \mathcal{H}^d \to \mathbb{R}$, the Fréchet derivative of $F$ is defined as follows.

**Definition 1.1** (Fréchet derivative in RKHS). *For a functional $F : \mathcal{H}^d \to \mathbb{R}$, its Fréchet derivative $DF[\psi]$ at $\psi \in \mathcal{H}^d$ is a function in $\mathcal{H}^d$ satisfying the following: For any $\xi \in \mathcal{H}^d$ with $\|\xi\|_{\mathcal{H}^d} < \infty$,*

$$\lim_{\epsilon \to 0} \frac{F[\psi + \epsilon\xi] - F[\psi]}{\epsilon} = \langle DF[\psi], \xi \rangle_{\mathcal{H}^d}.$$

Note that the Fréchet derivative at $\psi$, i.e. $DF[\psi]$, is a bounded linear operator from $\mathcal{H}^d$ to $\mathbb{R}$. It can be written in the form $DF[\psi](\xi) = \langle DF[\psi], \xi \rangle_{\mathcal{H}^d}$ due to the Riesz–Fréchet representation theorem.

## 1.1 Related Work on Functional Gradient Descent

A related functional gradient descent type method is the Stein Variation Gradient Descent (SVGD) method by Liu and Wang [2016]. SVGD considers the problem of minimizing the Kullback–Leibler (KL) divergence between a variable distribution and a posterior $p$. Note that SVGD updates the positions of a set of $N$ particles using the score function of the posterior $p$, i.e. $\nabla \log p$. Consequently, it requires the access to the target distribution function. Later, Liu [2017] prove that SVGD has convergence guarantee in its continuous-time limit (taking infinitesimal step size) using infinite number of particles ($N \to \infty$). In comparison, SD is designed to solve the significantly more complicated Sinkhon barycenter problem and has a stronger convergence guarantee. More precisely, while SD updates the measure using only a sampling machinery of the target measures (no score functions), it is guaranteed to converge sub-linearly to a stationary point when $\alpha$ is a *discrete* measure using *discrete* time steps. This is in sharp contrast to the results for SVGD.

In another work, Mroueh et al. [2019] considers minimizing the Maximum Mean Discrepancy (MMD) between a source measure and a variable measure. They solve this problem by incrementally following a Sobolev critic function and propose the Sobolev Descent (SoD) method. To show the global convergence of the measure sequence generated by SoD, Mroueh et al. [2019] assumes the *entire* sequence satisfies certain spectral properties, which is in general difficult to verify. Later, Arbel et al. [2019] consider the same MMD minimization problem from a gradient flow perspective. They propose two assumptions that if either one holds, the MMD gradient flow converges to the global solution. However, similar to [Mroueh et al., 2019], these assumptions have to be satisfied for the *entire* measure sequence. We note that the Sinkhorn barycenter is a strict generalization of the above MMD minimization problem and is hence much more challenging: By setting the number of source measures $n = 1$ and setting the entropy regularization parameter $\gamma = \infty$, problem (4) degenerates to the special case of MMD. Further, the MMD between two probability measures has a closed form expression while the Sinkhorn Divergence can only be described via a set of optimization problems. Consequently, the Sinkhorn barycenter is significantly more challenging. To guarantee global convergence, the proposed SD algorithm only requires one of accumulation points of the measure sequence to be fully supported on $\mathcal{X}$ with no restriction on the entire sequence.

## 2 Sinkhorn Barycenter

We first introduce the entropy-regularized optimal transport distance and its debiased version, a.k.a. the Sinkhorn divergence. Given two probability measures $\alpha, \beta \in \mathcal{M}_1^+(\mathcal{X})$, use $\Pi(\alpha, \beta)$ to denote the set of joint distributions over $\mathcal{X}^2$ with marginals $\alpha$ and $\beta$. For $\pi \in \Pi$, use $\langle c, \pi \rangle$ to denote the integral $\langle c, \pi \rangle = \int_{\mathcal{X}^2} c(x,y) \mathbf{d}\pi(x,y)$ and use $\mathrm{KL}(\pi \| \alpha \otimes \beta)$ to denote the Kullback-Leibler divergence between the candidate transport plan $\pi$ and the product measure $\alpha \otimes \beta$. The entropy-regularized optimal transport distance $\mathrm{OT}_\gamma(\alpha, \beta) : \mathcal{M}_1^+(\mathcal{X}) \times \mathcal{M}_1^+(\mathcal{X}) \to \mathbb{R}_+$ is defined as

$$\mathrm{OT}_\gamma(\alpha, \beta) = \min_{\pi \in \Pi(\alpha, \beta)} \langle c, \pi \rangle + \gamma \mathrm{KL}(\pi \| \alpha \otimes \beta). \tag{2}$$

Here, $\gamma > 0$ is a regularization parameter. Note that $\mathrm{OT}_\gamma(\alpha, \beta)$ is not a valid metric as there exists $\alpha \in \mathcal{M}_1^+(\mathcal{X})$ such that $\mathrm{OT}_\gamma(\alpha, \alpha) \neq 0$ when $\gamma \neq 0$. To remove this bias, Peyré et al. [2019] introduced the *Sinkhorn divergence* $\mathbb{S}_\gamma(\alpha, \beta) : \mathcal{M}_1^+(\mathcal{X}) \times \mathcal{M}_1^+(\mathcal{X}) \to \mathbb{R}_+$:

$$\mathbb{S}_\gamma(\alpha, \beta) := \mathrm{OT}_\gamma(\alpha, \beta) - \frac{1}{2}\mathrm{OT}_\gamma(\alpha, \alpha) - \frac{1}{2}\mathrm{OT}_\gamma(\beta, \beta), \tag{3}$$

which is a debiased version of $\mathrm{OT}_\gamma(\alpha, \beta)$. It is further proved that $\mathbb{S}_\gamma(\alpha, \beta)$ interpolates the Wasserstein distance and the Maximum Mean Discrepancy (MMD), and it is nonnegative, bi-convex and metrizes the convergence in law when the ground set $\mathcal{X}$ is compact and the metric $c$ is Lipschitz. Now given a set of probability measures $\{\beta_i\}_{i=1}^n$, the Sinkhorn barycenter is the measure $\alpha \in \mathcal{M}_1^+(\mathcal{X})$ that minimizes the average of Sinkhorn divergences

$$\min_{\alpha \in \mathcal{M}_1^+(\mathcal{X})} \left( \mathcal{S}_\gamma(\alpha) := \frac{1}{n} \sum_{i=1}^n \mathbb{S}_\gamma(\alpha, \beta_i) \right). \tag{4}$$

We will next focus on the properties of $\mathrm{OT}_\gamma$ since $\mathcal{S}_\gamma(\alpha)$ is the linear combination of these terms.

---

**Algorithm 1** `Sinkhorn Descent (SD)`

---

**Input:** measures $\{\beta_i\}_{i=1}^n$, a discrete initial measure $\alpha^0$, a step size $\eta$, and number of iterations $S$;
**Output:** A measure $\alpha^S$ that approximates the Sinkhorn barycenter of $\{\beta_i\}_{i=1}^n$;
**for** $t = 0$ to $S - 1$ **do**
$\quad \alpha^{t+1} := \mathcal{T}[\alpha^t]_\sharp \alpha^t$, with $\mathcal{T}[\alpha^t]$ defined in (11);
**end for**

---

**The Dual Formulation of** $\mathrm{OT}_\gamma$. As a convex program, the entropy-regularized optimal transport problem $\mathrm{OT}_\gamma$ (2) has a equivalent dual formulation, which is given as follows:

$$\mathrm{OT}_\gamma(\alpha, \beta) = \max_{f,g \in \mathcal{C}(\mathcal{X})} \langle f, \alpha \rangle + \langle g, \beta \rangle - \gamma \langle \exp((f \oplus g - c)/\gamma) - 1, \alpha \otimes \beta \rangle, \tag{5}$$

where we denote $[f \oplus g](x, y) = f(x) + g(y)$. The maximizers $f_{\alpha,\beta}$ and $g_{\alpha,\beta}$ of (5) are called the *Sinkhorn potentials* of $\mathrm{OT}_\gamma(\alpha, \beta)$. Define the Sinkhorn mapping $\mathcal{A} : \mathcal{C}(\mathcal{X}) \times \mathcal{M}_1^+(\mathcal{X}) \to \mathcal{C}(\mathcal{X})$ by

$$\mathcal{A}(f, \alpha)(y) = -\gamma \log \int_{\mathcal{X}} \exp\big((f(x) - c(x,y))/\gamma\big) \mathbf{d}\alpha(x). \tag{6}$$

The following lemma states the optimality condition for the Sinkhorn potentials $f_{\alpha,\beta}$ and $g_{\alpha,\beta}$.

**Lemma 2.1** (Optimality Peyré et al. [2019]). *The pair $(f, g)$ are the Sinkhorn potentials of the entropy-regularized optimal transport problem* (5) *if they satisfy*

$$f = \mathcal{A}(g, \beta), \alpha - a.e. \quad and \quad g = \mathcal{A}(f, \alpha), \beta - a.e.. \tag{7}$$

The Sinkhorn potential is the cornerstone of the entropy regularized OT problem. In the discrete case, it can be computed by a standard method in Genevay et al. [2016]. In particular, when $\alpha$ is discrete, $f$ can be simply represented by a finite dimensional vector since only its values on $\mathrm{supp}(\alpha)$ matter. We describe such method in Appendix A.1 for completeness. In the following, we treat the computation of Sinkhorn potentials as a blackbox, and refer to it as $\mathcal{SP}_\gamma(\alpha, \beta)$.

## 3 Methodology

We present the `Sinkhorn Descent (SD)` algorithm for the Sinkhorn barycenter problem (4) in two steps: We first reformulate (4) as an unconstrained functional minimization problem and then derive the descent direction as the negative functional gradient over a RKHS $\mathcal{H}^d$. Operating in RKHS allows us to measure the quality of the iterates using a so-called kernelized discrepancy which we introduce in Definition 4.1. This quantity will be crucial for our convergence analysis. The restriction of a functional optimization problem to RKHS is common in the literature as discussed in Remark 3.1.

**Alternative Formulation.** Instead of directly solving the Sinkhorn barycenter problem in the probability space $\mathcal{M}_1^+(\mathcal{X})$, we reformulate it as a functional minimization over all mappings on $\mathcal{X}$:

$$\min_{\mathcal{P}} \left( \mathcal{S}_\gamma(\mathcal{P}_\sharp \alpha_0) := \frac{1}{n} \sum_{i=1}^n \mathbb{S}_\gamma(\mathcal{P}_\sharp \alpha_0, \beta_i) \right), \tag{8}$$

where $\alpha_0 \in \mathcal{M}_1^+(\mathcal{X})$ is some given initial measure, and $\mathcal{P}_\sharp \alpha$ is the push-forward measure of $\alpha \in \mathcal{M}_1^+(\mathcal{X})$ under the mapping $\mathcal{P} : \mathcal{X} \to \mathcal{X}$. When $\alpha_0$ is sufficiently regular, e.g. absolutely continuous, for any $\alpha \in \mathcal{M}_1^+(\mathcal{X})$ there always exists a mapping $\mathcal{P}$ such that $\alpha = \mathcal{P}_\sharp \alpha_0$ (see Theorem 1.33 of [Ambrosio and Gigli, 2013]). Consequently, problems (8) and (4) are equivalent with appropriate initialization.

**Algorithm Derivation.** For a probability measure $\alpha$, define the functional $\mathcal{S}_\alpha : \mathcal{H}^d \to \mathbb{R}$

$$\mathcal{S}_\alpha[\psi] = \mathcal{S}_\gamma\big((\mathcal{I} + \psi)_\sharp \alpha\big), \psi \in \mathcal{H}^d. \tag{9}$$

Here $\mathcal{I}$ is the identity mapping and $\mathcal{S}_\gamma$ is defined in (4). Let $\alpha^t$ be the estimation of the Sinkhorn barycenter in the $t^{th}$ iteration. `Sinkhorn Descent (SD)` iteratively updates the measure $\alpha^{t+1}$ as

$$\alpha^{t+1} = \mathcal{T}[\alpha^t]_\sharp \alpha^t, \tag{10}$$

via the push-forward mapping (with $\eta > 0$ being a step-size)

$$\mathcal{T}[\alpha^t](x) = x - \eta \cdot D\mathcal{S}_{\alpha^t}[0](x). \tag{11}$$

Recall that $D\mathcal{S}_\alpha[0]$ is the Fréchet derivative of $\mathcal{S}_\alpha$ at $\psi = 0$ (see Definition 1.1). Note that $(\mathcal{I}+\psi)_\sharp \alpha = \alpha$ when $\psi = 0$. Our choice of the negative Fréchet derivative in $\mathcal{T}[\alpha^t]$ allows the objective $\mathcal{S}_\gamma(\alpha)$ to have the fastest descent at the current measure $\alpha = \alpha^t$. We our line the details of SD in Algorithm 1. Consequently, a solution of (8) will be found by finite-step compositions and then formally passing to the limit $\mathcal{P} = \lim_{t\to\infty} \left( \mathcal{P}^t := \mathcal{T}[\alpha^t] \circ \cdots \circ \mathcal{T}[\alpha^0] \right)$.

**Remark 3.1.** *We restrict $\psi$ in (9) to the space $\mathcal{H}^d$ to avoid the inherent difficulty when the perturbation of Sinkhorn potentials introduced by the mapping $(\mathcal{I} + \psi)$ can no longer be properly bounded (for $\psi \in \mathcal{H}^d$, we always have the upper bound (1) which is necessary in our convergence analysis). This restriction will potentially introduce error to the minimization of (8). However, this restriction is a common practice for general functional optimization problems: Both SVGD [Liu and Wang, 2016] and SoD [Mroueh et al., 2019] explicitly make such RKHS restriction on their transport mappings. [Arbel et al., 2019] constructs the transport mapping using the witness function of the Maximum Mean Discrepancy (MMD) which also lies in an RKHS.*

In what follows, we first derive a formula for the Fréchet derivative $D\mathcal{S}_{\alpha^t}[0]$ (see (13)) and then explain how it is efficiently computed. The proof of the next proposition requires additional continuity study of the Sinkhorn potentials and is deferred to Appendix C.5.

**Proposition 3.1.** *Recall the Fréchet derivative in Definition 1.1. Given $\alpha, \beta \in \mathcal{M}_1^+(\mathcal{X})$, for $\psi \in \mathcal{H}^d$ denote $F_1[\psi] = \mathrm{OT}_\gamma\big((\mathcal{I}+\psi)_\sharp\alpha, \beta\big)$ and $F_2[\psi] = \mathrm{OT}_\gamma\big((\mathcal{I}+\psi)_\sharp\alpha, (\mathcal{I}+\psi)_\sharp\alpha\big)$. Under Assumptions 4.1 and 4.2 (described below), we can compute*

$$DF_1[0](y) = \int_{\mathcal{X}} \nabla f_{\alpha,\beta}(x) k(x,y) \mathbf{d}\alpha(x), \quad DF_2[0](y) = 2 \int_{\mathcal{X}} \nabla f_{\alpha,\alpha}(x) k(x,y) \mathbf{d}\alpha(x), \tag{12}$$

*where $\nabla f_{\alpha,\beta}$ and $\nabla f_{\alpha,\alpha}$ are the gradients of the Sinkhorn potentials of $\mathrm{OT}_\gamma(\alpha,\beta)$ and $\mathrm{OT}_\gamma(\alpha,\alpha)$ respectively, and $k$ is the kernel function of the RKHS $\mathcal{H}$.*

Consequently the Fréchet derivative of the Sinkhorn Barycenter problem (9) can be computed by

$$D\mathcal{S}_\alpha[0](y) = \int_{\mathcal{X}} \frac{1}{n}\Big[\sum_{i=1}^n \nabla f_{\alpha,\beta_i}(x) - \nabla f_{\alpha,\alpha}(x)\Big] k(x,y) \mathbf{d}\alpha(x). \tag{13}$$

This quantity can be computed efficiently when $\alpha$ is discrete: Consider an individual term $\nabla f_{\alpha,\beta}$. Define $h(x,y) := \exp\left(\frac{1}{\gamma}(f_{\alpha,\beta}(x) + \mathcal{A}[f_{\alpha,\beta}, \alpha](y) - c(x,y))\right)$. Lemma 2.1 implies

$$\int h(x,y) \mathbf{d}\beta(y) = 1.$$

Taking derivative with respect to $x$ on both sides and rearranging terms, we have

$$\nabla f_{\alpha,\beta}(x) = \frac{\int_{\mathcal{X}} h(x,y)\nabla_x c(x,y) \mathbf{d}\beta(y)}{\int h(x,y) \mathbf{d}\beta(y)} = \int_{\mathcal{X}} h(x,y)\nabla_x c(x,y) \mathbf{d}\beta(y), \tag{14}$$

which itself is an expectation. Note that to evaluate (13), we only need $\nabla f_{\alpha,\beta}(x)$ on $\mathrm{supp}(\alpha)$. Using $\mathcal{SP}_\gamma(\alpha,\beta)$ (see the end of Section 2), the function value of $f_{\alpha,\beta}$ on $\mathrm{supp}(\alpha)$ can be efficiently computed. Together with the expression in (14), the gradients $\nabla f_{\alpha,\beta}(x)$ at $x \in \mathrm{supp}(\alpha)$ can also be obtained by a simple Monte-Carlo integration with respect to $\beta$.

## 4 Analysis

In this section, we analyze the finite time convergence and the mean field limit of SD under the following assumptions on the ground cost function $c$ and the kernel function $k$ of the RKHS $\mathcal{H}^d$.

**Assumption 4.1.** *The ground cost function $c(x,y)$ is bounded, i.e. $\forall x, y \in \mathcal{X}, c(x,y) \leq M_c$; $G_c$-Lipschitz continuous, i.e. $\forall x, x', y \in \mathcal{X}, |c(x,y) - c(x',y)| \leq G_c\|x - x'\|$; and $L_c$-Lipschitz smooth, i.e. $\forall x, x', y \in \mathcal{X}, \|\nabla_1 c(x,y) - \nabla_1 c(x',y)\| \leq L_c\|x - x'\|$.*

**Assumption 4.2.** *The kernel function $k(x,y)$ is bounded, i.e. $\forall x, y \in \mathcal{X}, k(x,y) \leq D_k$; $G_k$-Lipschitz continuous, i.e. $\forall x, x', y \in \mathcal{X}, |k(x,y) - k(x',y)| \leq G_c\|x - x'\|$.*

## 4.1 Finite Time Convergence Analysis

In this section, we prove that `Sinkhorn Descent` converges to a stationary point of problem (4) at the rate of $\mathcal{O}(\frac{1}{t})$, where $t$ is the number of iterations. We first introduce a discrepancy quantity.

**Definition 4.1.** *Recall the definition of the functional $\mathcal{S}_\alpha$ in (9) and the definition of Fréchet derivative in Definition 1.1. Given a probability measure $\alpha \in \mathcal{M}_1^+(\mathcal{X})$, the Kernelized Sinkhorn Barycenter Discrepancy (KSBD) for the Sinkhorn barycenter problem is defined as*

$$\mathbf{S}(\alpha, \{\beta_i\}_{i=1}^n) := \|D\mathcal{S}_\alpha[0]\|_{\mathcal{H}^d}^2. \tag{15}$$

Note that in each round $t$, $\mathbf{S}(\alpha^t, \{\beta_i\}_{i=1}^n)$ metrizes the stationarity of SD, which can be used to quantify the per-iteration improvement.

**Lemma 4.1** (Sufficient Descent)**.** *Recall the definition of the Sinkhorn Barycenter problem in (4) and the sequence of measures $\{\alpha^t\}_{t\geq 0}$ in (10) generated by SD (Algorithm 1). Under Assumption 4.1, if we have $\eta \leq \min\{1/(8L_f M_{\mathcal{H}}^2), 1/(8\sqrt{d}L_T M_{\mathcal{H}}^2)\}$, the Sinkhorn objective always decreases,*

$$\mathcal{S}_\gamma(\alpha_{t+1}) - \mathcal{S}_\gamma(\alpha_t) \leq -\eta/2 \cdot \mathbf{S}(\alpha^t, \{\beta_i\}_{i=1}^n). \tag{16}$$

*See $M_{\mathcal{H}}$ in (1), $L_f := 4G_c^2/\gamma + L_c$ and $L_T := 2G_c^2 \exp(3M_c/\gamma)/\gamma$[1].*

The proof of the lemma in given Appendix C.7. Based on this result, we can derive the following convergence result demonstrating that SD converges to a stationary point in a sublinear rate.

**Theorem 4.1** (Convergence)**.** *Suppose SD is initialized with $\alpha^0 \in \mathcal{M}_1^+(\mathcal{X})$ and outputs $\alpha^t \in \mathcal{M}_1^+(\mathcal{X})$ after $t$ iterations. Under Assumption 4.1, we have*

$$\min_t \mathbf{S}(\alpha^t, \{\beta_i\}_{i=1}^n) \leq 2\mathcal{S}_\gamma(\alpha^0)/(\eta t), \tag{17}$$

*where $0 < \eta \leq \min\{1/(8L_f M_{\mathcal{H}}^2), 1/(8\sqrt{d}L_T M_{\mathcal{H}}^2)\}$ is the step size.*

With a slight change to SD, we can conclude its last term convergence as elaborated in Appendix B.3.

**Remark 4.1** (Exponential dependence on $\frac{1}{\gamma}$)**.** *It is difficult to remove the exponential dependence on $\frac{1}{\gamma}$ in the above convergence result. Specifically, the term $\exp(1/\gamma)$ appears in bounding the smoothness of Sinkhorn potential, which is a key factor in bounding the sample complexity of the Sinkhorn divergence (see Theorem 2 and Lemma 3 of [Genevay et al., 2019a]). Consequently, the sample complexity in [Genevay et al., 2019a] can be improved if one manages to remove this factor. However, this would potentially violate the lower bound on the sample complexity of the hard-to-compute Wasserstein distance, which is the limit of the Sinkhorn divergence at $\gamma \to 0$.*
*Surprisingly, the empirical performance of SD does not suffer much from this factor: In our experiments (Section 5), to produce good visual results, we pick $\gamma = 10^{-4}$ and we still observe that SD quickly converges (even in the high dimensional Gaussian barycenter task).*

**Remark 4.2** (Implicit exponential dependence on the problem dimension)**.** *As shown in Lemma 4.1 and Theorem 4.1, our results depend on $\exp(M_c/\gamma)$ where $M_c$ is the upper bound on ground cost on the domain $\mathcal{X}$ which contains an implicit dependence on the problem dimension.*

## 4.2 Mean Field Limit Analysis

While `Sinkhorn Descent` accepts both discrete and continuous measures as initialization, in practice, we start from a discrete initial measure $\alpha_N^0$ with $|\text{supp}(\alpha_N^0)| = N$. If $\alpha_N^0$ is an empirical measure sampled from an underlying measure $\alpha_\infty^0$, we have the weak convergence at time $t = 0$, i.e. $\alpha_N^0 \rightharpoonup \alpha_\infty^0$ as $N \to \infty$. The mean field limit analysis demonstrates that `Sinkhorn Descent` preserves such weak convergence for any finite time $t$:

$$\alpha_N^0 \rightharpoonup \alpha_\infty^0 \Rightarrow \alpha_N^t = \text{SD}^t(\alpha_N^0) \rightharpoonup \alpha_\infty^t = \text{SD}^t(\alpha_\infty^0),$$

where we use $\text{SD}^t$ to denote the output of SD after $t$ steps and use $\rightharpoonup$ to denote the weak convergence.

**Lemma 4.2.** *Recall the push-forward mapping $\mathcal{T}[\alpha](x)$ in $\mathtt{SD}$ from (11) and recall $L_f$ in Lemma 4.1. Under Assumptions 4.1 and 4.2, for two probability measures $\alpha$ and $\alpha'$, we have*

$$d_{bl}(\mathcal{T}[\alpha]_{\sharp}\alpha, \mathcal{T}[\alpha']_{\sharp}\alpha') \leq (1 + \eta C) d_{bl}(\alpha, \alpha'), \tag{18}$$

*where $C = G_c G_k + \max\{dL_f D_k + dG_c G_k, D_k L_{bl}\}$ and $L_{bl} := 8G_c^2 \exp(6M_c/\gamma)$.*

The proof is presented in Appendix C.6. This is a discrete version of Dobrushin's estimate (section 1.4. in [Golse, 2016]). As a result, we directly have the following large N characterization of $\mathtt{SD}^t(\alpha_N^0)$.

**Theorem 4.2** (Mean Field Limit). *Let $\alpha_N^0$ be an empirical initial measure with $|\mathrm{supp}(\alpha_N^0)| = N$ and let $\alpha_\infty^0$ be the underlying measure such that $\alpha_N^0 \rightharpoonup \alpha_\infty^0$. Use $\mathtt{SD}^t(\alpha_N^0)$ and $\mathtt{SD}^t(\alpha_\infty^0)$ to denote the outputs of $\mathtt{SD}$ after $t$ iterations, under the initializations $\alpha_N^0$ and $\alpha_\infty^0$ respectively. Under Assumptions 4.1 and 4.2, for any finite time $t$, we have*

$$d_{bl}(\mathtt{SD}^t(\alpha_N^0), \mathtt{SD}^t(\alpha_\infty^0)) \leq (1 + \eta C)^t d_{bl}(\alpha_N^0, \alpha_\infty^0),$$

*and hence as $N \to \infty$ we have*

$$\alpha_N^t = \mathtt{SD}^t(\alpha_N^0) \rightharpoonup \alpha_\infty^t = \mathtt{SD}^t(\alpha_\infty^0). \tag{19}$$

### 4.3 KSBD as Discrepancy Measure

In this section, we show that, under additional assumptions, KSBD is a valid discrepancy measure, i.e. $\mathcal{S}_\gamma(\alpha) = 0$ implies that $\alpha$ is a global optimal solution to the Sinkhorn barycenter problem (4). The proof is provided in Appendix D. First, we introduce the following positivity condition.

**Definition 4.2.** *A kernel $k(x, x')$ is said to be integrally strictly positive definite (ISPD) w.r.t. a measure $\alpha \in \mathcal{M}_1^+(\mathcal{X})$, if $\forall \xi : \mathcal{X} \to \mathbb{R}^d$ with $0 < \int_{\mathcal{X}} \|\xi(x)\|^2 \mathbf{d}\alpha(x) < \infty$, it holds that*

$$\int_{\mathcal{X}^2} \xi(x) k(x, x') \xi(x') \mathbf{d}\alpha(x) \mathbf{d}\alpha(x') > 0. \tag{20}$$

**Theorem 4.3.** *Recall the Fréchet derivative of the Sinkhorn Barycenter problem in (13) and KSBD in (15). Denote $\xi(x) := \frac{1}{n}\sum_{i=1}^n \left(\nabla f_{\alpha,\beta_i}(x) - \nabla f_{\alpha,\alpha}(x)\right)$. We have $\int_{\mathcal{X}} \|\xi(x)\|^2 \mathbf{d}\alpha(x) < \infty$.*
*(i) If the kernel function $k(x, x')$ is ISPD w.r.t. $\alpha \in \mathcal{M}_1^+(\mathcal{X})$ and $\alpha$ is fully supported on $\mathcal{X}$, then the vanishing of KSBD, i.e. $\mathbf{S}(\alpha, \{\beta_i\}_{i=1}^n) = 0$, implies that $\alpha$ globally minimizes problem (4).*
*(ii) Use $\alpha^t$ to denote the output of $\mathtt{SD}$ after $t$ iterations. If further one of the accumulation points of the sequence $\{\alpha^t\}$ is fully supported on $\mathcal{X}$, then $\lim_{t \to \infty} \mathcal{S}_\gamma(\alpha^t) = \mathcal{S}_\gamma(\alpha^*)$.*

We show in Appendix D.2, under an absolutely continuous (a.c.) and fully supported (f.s.) initialization, $\alpha^t$ remains a.c. and f.s. for any finite $t$. This leads to our assumption in (ii): One of the accumulation points of $\{\alpha^t\}$ is f.s.. However, to rigorously analyze the support of $\alpha^t$ in the asymptotic case ($t \to \infty$) requires a separate proof. Establishing the global convergence of the functional gradient descent is known to be difficult in the literature, even for some much easier settings compared to our problem (4). For instance, [Mroueh et al., 2019, Arbel et al., 2019] prove the global convergence of their MMD descent algorithms. Both works require additional assumptions on the *entire* measure sequence $\{\alpha^t\}$ as detailed in Appendix D.3. See also the convergence analysis of SVGD in [Lu et al., 2019] under very strong assumptions of the score functions.

**Remark 4.3** (Initialization with finite particles). *In practice, we start with a sufficient number of particles. In this case, we cannot prove $\mathtt{SD}$ converges globally since (ii) of Theorem 4.3 does not hold. However, we observe that increasing the number of particles reduces the Sinkhorn divergence at convergence, but with diminishing returns.*

## 5 Experiments

We conduct experimental studies to show the efficiency and efficacy of $\mathtt{Sinkhorn\ Descent}$ by comparing with the recently proposed functional Frank-Wolfe method ($\mathtt{FW}$) from [Luise et al., 2019][2]. Note that in round $t$, $\mathtt{FW}$ requires to *globally* minimize the nonconvex function

$$Q(x) := \sum_{i=1}^n f_{\alpha^t, \beta_i}(x) - f_{\alpha^t, \alpha^t}(x), \tag{21}$$

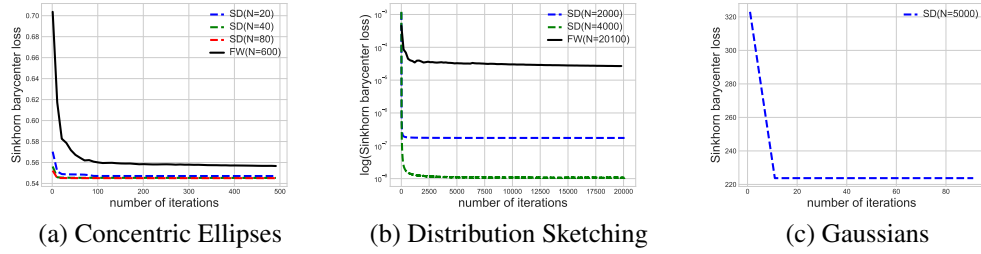

| (a) Concentric Ellipses | (b) Distribution Sketching | (c) Gaussians |

Figure 1: $N$ is the support size. FW is not included in (c) as it is impractical in high-dimensional problems (here, the dimension is 100)

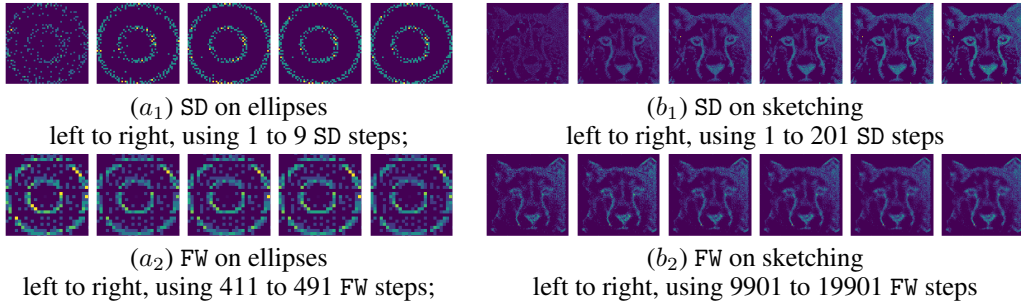

$(a_1)$ SD on ellipses
left to right, using 1 to 9 SD steps;

$(b_1)$ SD on sketching
left to right, using 1 to 201 SD steps

$(a_2)$ FW on ellipses
left to right, using 411 to 491 FW steps;

$(b_2)$ FW on sketching
left to right, using 9901 to 19901 FW steps

Figure 2: Visual results of the ellipses and sketching problem.

in order to choose the next Dirac measure to be added to the support. Here, $f_{\alpha^t,\beta_i}$ and $f_{\alpha^t,\alpha^t}$ are the Sinkhorn potentials. Such operation is implemented by an exhaustive grid search so that FW returns a reasonably accurate solution. Consequently, FW is computationally expensive even for low dimensional problems and we only compare SD with FW in the first two image experiments, where $d = 2$. (the grid size used in FW grows exponentially with $d$.)

Importantly, the size of the support $N$ affects the computational efficiency as well as the solution quality of both methods. A large support size usually means higher computational complexity but allows a more accurate approximation of the barycenter. However, since SD and FW have different support size patterns, it is hard to compare them directly: The support size of SD is fixed after its initialization while FW starts from an initial small-size support and gradually increases it during the optimization procedure. We hence fix the support size of the output measure from FW and vary the support size of SD for a more comprehensive comparison. In the following, the entropy regularization parameter $\gamma$ is set to $10^{-4}$ in all tasks to produce results of good visual quality.

**Barycenter of Concentric Ellipses** We compute the barycenter of 30 randomly generated concentric ellipses similarly as done in [Cuturi and Doucet, 2014, Luise et al., 2019]. We run FW for 500 iterations and hence the output measure of FW has support size $N = 600$ (FW increases its support size by 1 in each iteration). SD is initialized with a discrete uniform distribution with support size varying from $N \in \{20, 40, 80\}$. Note that in these experiments the chosen support size for SD is even smaller than the initial support size of FW. The result is reported in Figure 1(a). In terms of convergence rate, we observe that SD is much faster than FW. Even 20 iterations are sufficient for SD to find a good solution. More importantly, in terms of the quality of the solution, SD with support size $N = 20$ outperforms FW with final support size $N = 600$. In fact, FW cannot find a solution with better quality even with a larger support size. This phenomenon is due to an inevitable limitation of the FW optimization procedure: Each FW step requires to globally minimize the non-convex function (32) via an exhaustive grid search. This introduces an inherent error to the procedure as the actual solution to (32) potentially resides outside the grid points. Such error limits the accuracy of FW even when the number of particles grows. In contrast, SD adjusts the particles to minimize the objective without any inherent error. As a result, we observe SD outperforms FW on both efficiency and accuracy.

**Distribution Sketching** We consider a special case of the barycenter problem where we only have one source distribution, similarly as done in [Luise et al., 2019]. This problem can be viewed as

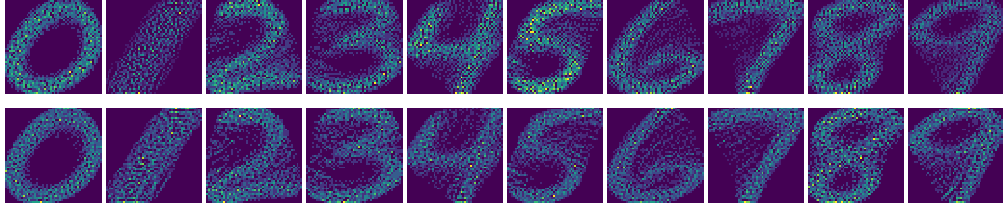

Figure 3: Visual results on MNIST. The first row contains results from SD and the second row contains results from free-support method in [Cuturi and Doucet, 2014].

approximating a given distribution with a fixed support size budget and is hence called distribution sketching. Specifically, a natural image of a cheetah is used as the source measure in $\mathbb{R}^2$. We run FW for 20000 iterations and the support size of SD is $N \in \{2000, 4000\}$. The result is reported in Figure 1(b). Since we only have one source measure, the Sinkhorn barycenter loss is very small and hence we use log-scale in the y-axis. We can observe that SD outperforms FW in terms of the quality of the solution as well as the convergence rate.

**Barycenter of Gaussians**   To demonstrate the efficiency of SD on high dimensional problems, we consider the problem of finding the barycenter of multivariate Gaussian distributions. Concretely, we pick 5 isotropic Gaussians in $\mathbb{R}^{100}$ with different means. For each of them, we sample an empirical measure with 50000 points and used the obtained empirical measures as source measures. We initialize SD with an empirical measure sampled from the uniform distribution with support size $N = 5000$. We did not compare with FW as the global minimizer of $Q(x)$ can not be computed in $\mathbb{R}^{100}$. The result is reported in Figure 1(c). We can see that just like the previous two experiments, SD converges in less than 20 iterations.

**Visual Results on Ellipses and Sketching.**   To compare SD with FW visually, we allow SD with FW to have a similar amount of particles in the ellipses and sketching tasks, and report the results in Figure 2. Specifically, in $(a_1)$ SD has 500 particles while in $(a_2)$ FW has 511 to 591 particles (recall that the support size of FW grows over iterations); in $(b_1)$ SD has 8000 particles while in $(a_2)$ FW has 10001 to 20001 particles. In all cases FW has at least as much particles as SD does while having significantly more steps. However, the visual result produced by SD is clearly better than FW: in $(a_1)$, the circle is very clear in the last picture while in $(a_2)$ all pictures remain vague; in $(b_1)$, the eyes of cheetah are clear, but in $(b_2)$ the eyes remain gloomy.

**Additional Visual Results on MNIST**   We provide additional results on the MNIST dataset. In this problem, we view the (low resolution) images of every individual digit (0-9) as one of target distributions and our goal is to compute their (high resolution) Sinkhorn barycenter (we randomly select 100 images of each digit in MNIST). We compare SD with the free-support method in [Cuturi and Doucet, 2014] (we directly use the implementation from the PythonOT library[3]). Both methods use 2500 particles and are run for 100 iterations. In the 2D-histogram image of Figure 3, brighter pixels means more particles in a local region. We can observe that the particles of SD are more concentrated on the digits compared to the ones of [Cuturi and Doucet, 2014]. We only report the visual results but do not compare the exact barycenter loss since [Claici et al., 2018] considers the exact Wasserstein barycenter problem with no entropy regularization. We do not compare with [Chewi et al., 2020] since it only applies to the barycenter problem of Gaussian distributions.

## 6   Broader Impact

This work has the following potential positive impact in the society: We propose the first algorithm for the Sinkhorn barycenter problem that is scalable with respect to the problem dimension $d$ (linear dependence), while existing works all have an exponential dependence on $d$. Further, we expect that this functional gradient descent method can be applied to more general optimization problems involving distribution sampling: In principle, the negative gradient of the dual variables instructs the particles in the measure to search the landscape of the minimizer.

## Acknowledgment

This work is supported by NSF CPS-1837253.

## Footnotes

[1]We acknowledge the factor $\exp(1/\gamma)$ is non-ideal, but such quantity constantly appears in the literature related to the Sinkhorn divergence, e.g. Theorem 5 in [Luise et al., 2019] and Theorem 3 in [Genevay et al., 2019a]. It would be an interesting future work to remove this factor.

[2][Claici et al., 2018] is not included as it only applies to the Wasserstein barycenter problem ($\gamma = 0$).

[3]`https://pythonot.github.io/index.html`

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
