[Supplementary Material]

# A Preliminaries on the Sinkhorn Potentials

**Lemma A.1** (Lemma A.2 elaborated). *For a probability measure $\alpha \in \mathcal{M}_1^+(\mathcal{X})$, use $\alpha-a.e.$ to denote "almost everywhere w.r.t. $\alpha$". The pair $(f, g)$ are the Sinkhorn potentials of the entropy-regularized optimal transport problem* (5) *if they satisfy*

$$f = \mathcal{A}(g, \beta), \alpha - a.e. \quad and \quad g = \mathcal{A}(f, \alpha), \beta - a.e., \tag{22}$$

*or equivalently*

$$\int_{\mathcal{X}} h(x,y)\mathbf{d}\beta(y) = 1, \ \alpha - a.e., \tag{23}$$

$$\int_{\mathcal{X}} h(x,y)\mathbf{d}\alpha(x) = 1, \ \beta - a.e., \tag{24}$$

*where $h(x,y) := \exp\left(\frac{1}{\gamma}(f(x) + g(y) - c(x,y))\right)$.*

One can observe that the Sinkhorn potentials are not unique. In fact, for $\alpha \neq \beta$, the pair $(f_{\alpha,\beta}, g_{\alpha,\beta})$ remains optimal under a constant shift, i.e. $(f_{\alpha,\beta} + C, g_{\alpha,\beta} - C)$ are still the Sinkhorn potentials of $\mathrm{OT}_\gamma(\alpha, \beta)$ for an arbitrary finite $C \in \mathbb{R}$. Fortunately, it is proved in Cuturi [2013] that the Sinkhorn potentials are unique up to such scalar translation.
To reduce the ambiguity, we fix an $x_o \in \mathcal{X}$ and choose $f_{\alpha,\beta}(x_o) = 0$, since otherwise we can always shift $f_{\alpha,\beta}$ and $g_{\alpha,\beta}$ by the amount of $f_{\alpha,\beta}(x_o)$. While it is possible that $x_o \notin \mathrm{supp}(\alpha)$, such choice of $f_{\alpha,\beta}$ is still feasible. This is because the Sinkhorn potentials can be naturally extended to the entire $\mathcal{X}$ from Lemma 2.1, even though the above optimality condition characterizes the Sinkhorn potentials on $\mathrm{supp}(\alpha), \mathrm{supp}(\beta)$ only.
Further, this choice of $f_{\alpha,\beta}$ allows us to bound $\|f_{\alpha,\beta}\|_\infty$ given that the ground cost function $c$ is bounded on $\mathcal{X}$.

**Assumption A.1.** *The cost function $c(x,y)$ is bounded: $\forall x, y \in \mathcal{X}, c(x,y) \leq M_c$.*

**Lemma A.2** (Boundedness of the Sinkhorn Potentials). *Let $(f, g)$ be the Sinkhorn potentials of problem* (5) *and assume that there exists $x_o \in \mathcal{X}$ such that $f(x_o) = 0$ (otherwise shift the pair by $f(x_o)$). Then, under Assumption A.1, $\|f\|_\infty \leq 2M_c$ and $\|g\|_\infty \leq 2M_c$.*

Next, we analyze the Lipschitz continuity of the Sinkhorn potential $f_{\alpha,\beta}(x)$ with respect to $x$.

**Assumption A.2.** *The cost function $c$ is $G_c$-Lipschitz continuous with respect to one of its inputs:*

$$\forall x, x' \in \mathcal{X}, |c(x,y) - c(x',y)| \leq G_c\|x - x'\|.$$

Assumption A.2 implies that $\nabla_x c(x,y)$ exists and for all $x, y \in \mathcal{X}, \|\nabla_x c(x,y)\| \leq G_c$. It further ensures the Lipschitz-continuity of the Sinkhorn potential.

**Lemma A.3** (Proposition 12 of Feydy et al. [2019]). *Under Assumption A.2, for a fixed pair of measures $(\alpha, \beta)$, the Sinkhorn potential $f_{\alpha,\beta} : \mathcal{X} \to \mathbb{R}$ is $G_c$-Lipschitz continuous,*

$$\forall x, x' \in \mathcal{X}, |f_{\alpha,\beta}(x) - f_{\alpha,\beta}(x')| \leq G_c\|x - x'\|. \tag{25}$$

*Further, the gradient $\nabla f_{\alpha,\beta}$ exists at every point $x \in \mathcal{X}$, and $\|\nabla f_{\alpha,\beta}(x)\| \leq G_c, \forall x \in \mathcal{X}$.*

**Assumption A.3.** *The gradient of the cost function $c$ is $L_c$-Lipschitz continuous: for all $x, x' \in \mathcal{X}$,*

$$\|\nabla_1 c(x,y) - \nabla_1 c(x',y)\| \leq L_c\|x - x'\|.$$

**Lemma A.4.** *Assume Assumptions A.2 and A.3, and denote $L_f := 4G_c^2/\gamma + L_c$. For a pair of measures $(\alpha, \beta)$, the gradient of the corresponding Sinkhorn potential $f_{\alpha,\beta} : \mathcal{X} \to \mathbb{R}$ is Lipschitz continuous,*

$$\forall x, x' \in \mathcal{X}, \|\nabla f_{\alpha,\beta}(x) - \nabla f_{\alpha,\beta}(x')\| \leq L_f\|x - x'\|. \tag{26}$$

## A.1 Computation of Sinkhorn Potentials

The Sinkhorn potential is the cornerstone of the entropy regularized OT problem $\mathrm{OT}_\gamma(\alpha, \beta)$. Hence, a key component of our method is to efficiently compute this quantity. An efficient method is given in Genevay et al. [2016] when both $\alpha$ and $\beta$ are discrete measures (discrete case), as well as when $\alpha$

is discrete but $\beta$ is continuous (semi-discrete case). More precisely, by plugging in the optimality condition on $g$ in (7), the dual problem (5) becomes

$$\mathrm{OT}_\gamma(\alpha, \beta) = \max_{f \in \mathcal{C}} \langle f, \alpha \rangle + \langle \mathcal{A}(f, \alpha), \beta \rangle. \tag{27}$$

Note that (27) only depends on the values of $f$ on the support of $\alpha$, $\mathrm{supp}(\alpha)$, which can be represented by a finite dimensional vector $\mathbf{f} \in \mathbb{R}^{|\mathrm{supp}(\alpha)|}$. Viewing the discrete measure $\alpha$ as a weight vector $\omega_\alpha$ on $\mathrm{supp}(\alpha)$, we have

$$\mathrm{OT}_\gamma(\alpha, \beta) = \max_{\mathbf{f} \in \mathbb{R}^d} \left\{ F(\mathbf{f}) := \mathbf{f}^\top \omega_\alpha + \mathbb{E}_{y \sim \beta} \left[ \mathcal{A}(\mathbf{f}, \alpha)(y) \right] \right\},$$

that is, $\mathrm{OT}_\gamma(\alpha, \beta)$ is equivalent to a standard concave stochastic optimization problem, where randomness of the problem comes from $\beta$ (see Proposition 2.1 in Genevay et al. [2016]). Hence, the problem can be solved using off-the-shelf stochastic optimization methods, e.g. stochastic gradient descent (SGD). Since the entropy regularized optimal transport problem is strongly convex, SGD converges at the rate $\mathcal{O}(1/k)$ for such problem, where $k$ is the number of SGD steps. Besides, the per-iteration complexity is $O(n^2)$ where $n$ is the support size of input measures. In the main body, this method is referred as $\mathcal{SP}_\gamma(\alpha, \beta)$.

## B  Lipschitz Continuity of the Sinkhorn Potential

In this section, we provide several lemmas to show the Lipschitz continuity (w.r.t. the underlying probability measures) of the Sinkhorn potentials and the functional gradients we derived in Proposition 3.1. These lemmas will be used in the convergence analysis and the mean field analysis for SD .

### B.1  Lipschitz Continuity Study: Sinkhorn Potentials

We first show the Lipschitz continuity of the Sinkhorn potential w.r.t. the bounded Lipschitz norm of the input measures. The bounded Lipschitz metric of measures $d_{bl} : \mathcal{M}_1^+(\mathcal{X}) \times \mathcal{M}_1^+(\mathcal{X}) \to \mathbb{R}_+$ with respect to the bounded continuous test functions is defined as

$$d_{bl}(\alpha, \beta) := \sup_{\|\xi\|_{bl} \leq 1} |\langle \xi, \alpha \rangle - \langle \xi, \beta \rangle|,$$

where, given a function $\xi \in \mathcal{C}(\mathcal{X})$, we denote

$$\|\xi\|_{bl} := \max\{\|\xi\|_\infty, \|\xi\|_{lip}\}, \quad \text{with} \|\xi\|_{lip} := \max_{x, y \in \mathcal{X}} \frac{|\xi(x) - \xi(y)|}{\|x - y\|}.$$

We note that $d_{bl}$ metrizes the weak convergence of probability measures (see Theorem 1.12.4 in Van Der Vaart and Wellner [1996]), i.e. for a sequence of probability measures $\{\alpha_n\}$,

$$\lim_{n \to \infty} d_{bl}(\alpha_n, \alpha) = 0 \Leftrightarrow \alpha_n \rightharpoonup \alpha.$$

**Lemma B.1.** *(i) Under Assumptions A.1 and A.2, for two given pairs of measures $(\alpha, \beta)$ and $(\alpha', \beta')$, the Sinkhorn potentials are Lipschitz continuous with respect to the bounded Lipschitz metric:*

$$\|f_{\alpha, \beta} - f_{\alpha', \beta'}\|_\infty \leq G_{bl}[d_{bl}(\alpha', \alpha) + d_{bl}(\beta', \beta)],$$
$$\|g_{\alpha, \beta} - g_{\alpha', \beta'}\|_\infty \leq G_{bl}[d_{bl}(\alpha', \alpha) + d_{bl}(\beta', \beta)].$$

*where $G_{bl} = 2\gamma \exp(2M_c/\gamma) G'_{bl} / (1 - \lambda^2)$ with $G'_{bl} = \max\{\exp(3M_c/\gamma), 2G_c \exp(3M_c/\gamma)/\gamma\}$ and $\lambda = \frac{\exp(M_c/\gamma) - 1}{\exp(M_c/\gamma) + 1}$.*
*(ii) If $(\alpha', \beta')$ are of the particular form $\alpha' = T_{\phi\sharp}\alpha$ and $\beta' = \beta$ where $T_\phi(x) = x + \phi(x), \phi \in \mathcal{H}^d$, we further have that the Sinkhorn potentials are Lipschitz continuous with respect to the mapping $\phi$. That is, letting $G_T := 2G_c \exp(3M_c/\gamma)/\gamma$ and $\epsilon > 0$, we have*

$$\|f_{T_\sharp\alpha, \beta} - f_{\alpha, \beta}\|_\infty \leq G_T \|\phi\|_{2, \infty},$$
$$\|g_{T_\sharp\alpha, \beta} - g_{\alpha, \beta}\|_\infty \leq G_T \|\phi\|_{2, \infty}.$$

Please see the proof in Appendix C.3. Importantly, this lemma implies that the weak convergence of $(\alpha, \beta)$ ensures the convergence of the Sinkhorn potential: $(\alpha', \beta') \rightharpoonup (\alpha, \beta) \Rightarrow (f_{\alpha', \beta'} \to f_{\alpha, \beta})$ in terms of the $L^\infty$ norm.

**Remark B.1.** *While we acknowledge that the factor $\exp 1/\gamma$ is non-ideal, such quantity constantly appears in the literature related to the Sinkhorn divergence, e.g. Theorem 5 in Luise et al. [2019] and Theorem 3 in Genevay et al. [2019b]. It would be an interesting future work to remove this factor.*

**Remark B.2.** *We note that the Lemma B.1 is strictly stronger than preexisting results: (1) Proposition 13 of Feydy et al. [2019] only shows that the dual potentials are continuous (not Lipschitz continuous) with the input measures, which is insufficient for the mean field limit analysis conducted in Section 4.2. (2) Under the infinity norm $\|\cdot\|_\infty$, Luise et al. [2019] bound the variation of the Sinkhorn potential by the total variation distance of probability measures $(\alpha, \beta)$ and $(\alpha', \beta')$. Such result means that strong convergence of $(\alpha, \beta)$ implies the convergence of the corresponding Sinkhorn potential. This is strictly weaker than (i) of Lemma B.1. (3) Further, to prove the weak convergence of the corresponding Sinkhorn potential, Proposition E.5 of the above work Luise et al. [2019] requires the cost function $c \in \mathcal{C}^{s+1}$ with $s > d/2$, where $d$ is the problem dimension. However, Lemma B.1 only assumes $c \in \mathcal{C}^1$, independent of $d$. Hence, Lemma B.1 makes a good contribution over existing results.*

The continuity results in Lemma B.1 can be further extended to the gradient of the Sinkhorn potentials.

**Lemma B.2.** *(i) Under Assumptions A.1 and A.2, for two given pairs of measures $(\alpha, \beta)$ and $(\alpha', \beta')$, with $G_{bl}[d_{bl}(\alpha', \alpha) + d_{bl}(\beta', \beta)] \leq 1$, the gradient of the Sinkhorn potentials are locally Lipschitz continuous with respect to the bounded Lipschitz metric: With $L_{bl} = 2G_c G_{bl}$,*

$$\|\nabla f_{\alpha,\beta} - \nabla f_{\alpha',\beta'}\|_\infty \leq L_{bl}[d_{bl}(\alpha', \alpha) + d_{bl}(\beta', \beta)],$$
$$\|\nabla g_{\alpha,\beta} - \nabla g_{\alpha',\beta'}\|_\infty \leq L_{bl}[d_{bl}(\alpha', \alpha) + d_{bl}(\beta', \beta)].$$

*(ii) If $(\alpha', \beta')$ are of the particular form $\alpha' = T_{\phi\sharp}\alpha$ and $\beta' = \beta$ where $T_\phi(x) = x + \phi(x)$ for $\phi \in \mathcal{H}^d$, we further have that the Sinkhorn potentials are Lipschitz continuous with respect to the mapping $\phi$: Let $G_T := 2G_c \exp(3M_c/\gamma)/\gamma$ and assume $2G_T \|\phi\|_{2,\infty} \leq 1$. We have with $L_T = 2G_c G_T$*

$$\|\nabla f_{T_\sharp\alpha,\beta} - \nabla f_{\alpha,\beta}\|_\infty \leq L_T \|\phi\|_{2,\infty},$$
$$\|\nabla g_{T_\sharp\alpha,\beta} - \nabla g_{\alpha,\beta}\|_\infty \leq L_T \|\phi\|_{2,\infty}.$$

The proof is given in Appendix C.4. The two lemmas B.1 B.2 are crucial to the analysis of the finite-time convergence and the mean field limit of `Sinkhorn Descent` .

## B.2 Lipschitz Continuity Study: Fréchet Derivative

From Definition 1.1, the Fréchet derivatives derived in Proposition 3.1 are functions in $\mathcal{H}^d$ mapping from $\mathcal{X}$ to $\mathbb{R}^d$. They are Lipschitz continuous provided that the kernel function $k$ is Lipschitz.

**Assumption B.1.** *The kernel function $k : \mathcal{X} \times \mathcal{X} \to \mathbb{R}_+$ is Lipschitz continuous on $\mathcal{X}$: for any $y$ and $x, x' \in \mathcal{X}$*

$$|k(x, y) - k(x', y)| \leq G_k \|x - x'\|. \tag{28}$$

**Lemma B.3.** *Define the functional on RKHS $F[\psi] := \mathrm{OT}_\gamma \big( (\mathcal{I} + \psi)_\sharp \alpha, \beta \big)$. Assume Assumptions A.1, A.2, A.3, and B.1. The Fréchet derivative $DF[0] \in \mathcal{H}^d$ is Lipschitz continuous: Denote $L_\psi = G_c G_k$. For any $x, x' \in \mathcal{X}$,*

$$\|DF[0](x) - DF[0](x')\| \leq L_\psi \|x - x'\|.$$

Using the above result, the functional gradient (13) can be shown to be Lipschitz continuous.

**Corollary B.1.** *Assume Assumptions A.1, A.2, A.3, and B.1. Recall $L_\psi = G_c G_k$ from the above lemma. The Fréchet derivative $DS_\alpha[0] \in \mathcal{H}^d$ is Lipschitz continuous: For any $x, x' \in \mathcal{X}$,*

$$\|DS_\alpha[0](x) - DS_\alpha[0](x')\| \leq L_\psi \|x - x'\|.$$

## B.3 Last term convergence of `SD`

With a slight change to `SD` , we can claim its last term convergence: In each iteration, check if $\mathbf{S}(\alpha^t, \{\beta_i\}_{i=1}^n) \leq \epsilon$. If it holds, then we have already identified an $\epsilon$ approximate stationary point and we terminate `SD` ; otherwise we proceed. The termination happens within $\mathcal{O}(1/\epsilon)$ loops as the nonnegative objective (4) is reduced at least $\mathcal{O}(\epsilon)$ per-round.

# C Proof of Lemmas

## C.1 Proof of Lemma A.3

For simplicity, we omit the subscript of the Sinkhorn potential $f_{\alpha,\beta}$ and simply use $f$. Recall the definition of $h(x, y)$ in Lemma A.1:

$$h(x, y) = \exp\left(\frac{1}{\gamma}(f(x) + g(y) - c(x, y))\right).$$

Subtract the optimality condition (23) at different points $x$ and $x'$ to derive

$$\int_{\mathcal{X}} \left(h(x, y) - h(x', y)\right)\mathbf{d}\beta(y) = 0 \Rightarrow$$

$$\int_{\mathcal{X}} h(x', y)\left(\exp\left(\frac{f(x) - f(x') - c(x, y) + c(x', y)}{\gamma}\right) - 1\right)\mathbf{d}\beta(y) = 0$$

Since $\int_{\mathcal{X}} h(x', y)\mathbf{d}\beta(y) = 1$ (Lemma A.1), we have

$$\int_{\mathcal{X}} h(x', y)\exp\left(\frac{f(x) - f(x') - (c(x, y) - c(x', y))}{\gamma}\right)\mathbf{d}\beta(y) = 1$$

$$\Rightarrow \int_{\mathcal{X}} h(x', y)\exp\left(\frac{c(x', y) - c(x, y)}{\gamma}\right)\mathbf{d}\beta(y) = \exp\left(\frac{f(x') - f(x)}{\gamma}\right).$$

Further, since we have $h(x', y) \geq 0$ and from Assumption A.1 we have

$$\exp\left(\frac{c(x', y) - c(x, y)}{\gamma}\right) \leq \exp\left(\frac{|c(x', y) - c(x, y)|}{\gamma}\right) \leq \exp\left(\frac{G_c\|x' - x\|}{\gamma}\right),$$

we derive

$$\left|\frac{f(x') - f(x)}{\gamma}\right| \leq \left|\log\left(\int_{\mathcal{X}} h(x', y)\exp\left(\frac{G_c\|x' - x\|}{\gamma}\right)\mathbf{d}\beta(y)\right)\right| \leq \frac{G_c\|x' - x\|}{\gamma},$$

by using $\int_{\mathcal{X}} h(x', y)\mathbf{d}\beta(y) = 1$ again, which consequently leads to

$$|f(x') - f(x)| \leq G_c\|x' - x\|.$$

## C.2 Proof of Lemma A.4

Recall the expression of $\nabla f$ in (14):

$$\nabla f(x) = \int_{\mathcal{X}} h(x, y)\nabla_x c(x, y)\mathbf{d}\beta(y), \tag{29}$$

where $h(x, y) := \exp\left(\frac{1}{\gamma}(f_{\alpha,\beta}(x) + \mathcal{A}[f_{\alpha,\beta}, \alpha](y) - c(x, y))\right)$. For any $x, x' \in \mathcal{X}$ such that $\|x_1 - x_2\| \leq \frac{\gamma}{2G_c}$, we bound

$$\|\nabla f(x) - \nabla f(x')\| = \left\|\int_{\mathcal{X}} h(x, y)\nabla_x c(x, y) - h(x', y)\nabla_x c(x', y)\mathbf{d}\beta(y)\right\|$$

$$\leq \int_{\mathcal{X}} \|h(x, y)\nabla_x c(x, y) - h(x', y)\nabla_x c(x', y)\|\mathbf{d}\beta(y)$$

To bound the last integral, observe that

$$h(x, y)\nabla_x c(x, y) - h(x', y)\nabla_x c(x', y)$$
$$= h(x, y)\left(\nabla_x c(x, y) - \nabla_x c(x', y)\right) + \left(h(x, y) - h(x', y)\right)\nabla_x c(x', y),$$

and therefore

$$\|h(x, y)\nabla_x c(x, y) - h(x', y)\nabla_x c(x', y)\|$$
$$\leq h(x, y)\|\nabla_x c(x, y) - \nabla_x c(x', y)\| + |h(x, y) - h(x', y)|\|\nabla_x c(x', y)\|.$$

For the first term, we use the Lipschitz continuity of $\nabla_x c$ from Assumption A.3 to bound

$$h(x,y)\|\nabla_x c(x,y) - \nabla_x c(x',y)\| \leq L_c h(x,y)\|x - x'\|.$$

For the second term, observe that $\|\nabla_x c(x',y)\| \leq G_c$ from Assumption A.2 and

$$|h(x,y) - h(x',y)| = h(x',y)|\exp(\frac{f(x) - f(x') - c(x,y) + c(x',y)}{\gamma}) - 1|$$

$$< 2h(x',y)|\frac{f(x) - f(x') - c(x,y) + c(x',y)}{\gamma}|.$$

Since $|\exp(z) - 1| < 2|z|$ when $|z| \leq 1$ ($z = |\frac{f(x)-f(x')-c(x,y)+c(x',y)}{\gamma}| \leq 1$ from the restriction on $\|x - x'\|$), we further derive

$$|h(x,y) - h(x',y)| \leq \frac{2G_c}{\gamma} h(x',y)[2G_c\|x - x'\|] = \frac{4G_c^2}{\gamma} h(x',y)\|x - x'\|.$$

Using the optimality condition $\int_{\mathcal{X}} h(x',y)\mathbf{d}\beta(y) = 1$ and $\int_{\mathcal{X}} h(x,y)\mathbf{d}\beta(y) = 1$ from Lemma 2.1, we derive

$$\|\nabla f(x) - \nabla f(x')\| \leq \int_{\mathcal{X}} L_c h(x,y)\|x-x'\| + \frac{4G_c^2}{\gamma} h(x',y)\|x-x'\|\mathbf{d}\beta(y) = (L_c + \frac{4G_c^2}{\gamma})\|x-x'\|.$$

This implies that $\nabla^2 f(x)$ exists and is bounded from above: $\forall x \in \mathcal{X}, \|\nabla^2 f(x)\| \leq L_f$, which concludes the proof.

### C.3 Proof of Lemma B.1

Let $(f,g)$ and $(f',g')$ be the Sinkhorn potentials to $\mathrm{OT}_\gamma(\alpha,\beta)$ and $\mathrm{OT}_\gamma(\alpha',\beta')$ respectively. Denote $u := \exp(f/\gamma)$, $v := \exp(g/\gamma)$ and $u' := \exp(f'/\gamma)$, $v' := \exp(g'/\gamma)$. From Lemma A.2, $u$ is bounded in terms of the $L^\infty$ norm:

$$\|u\|_\infty = \max_{x\in\mathcal{X}} |u(x)| = \max_{x\in\mathcal{X}} \exp(f/\gamma) \leq \exp(2M_c/\gamma),$$

which also holds for $v, u', v'$. Additionally, from Lemma A.3, $\nabla u$ exists and $\|\nabla u\|$ is bounded:

$$\max_x \|\nabla u(x)\| = \max_x \frac{1}{\gamma}|u(x)|\|\nabla f(x)\| \leq \frac{1}{\gamma}\|u(x)\|_\infty \max_x \|\nabla f(x)\| \leq G_c \exp(2M_c/\gamma)/\gamma.$$

Define the mapping $A_\alpha \mu := 1/(L_\alpha \mu)$ with

$$L_\alpha \mu = \int_{\mathcal{X}} l(\cdot,y)\mu(y)\mathbf{d}\alpha(y),$$

where $l(x,y) := \exp(-c(x,y)/\gamma)$. From Assumption A.1, we have $\|l\|_\infty \leq \exp(M_c/\gamma)$ and from Assumption A.2 we have $\|\nabla_x l(x,y)\| \leq \exp(M_c/\gamma)\frac{G_c}{\gamma}$. From the optimality condition of $f$ and $g$, we have $v = A_\alpha u$ and $u = A_\beta v$. Similarly, $v' = A_{\alpha'} u'$ and $u' = A_{\beta'} v'$. Further use $d_H : \mathcal{C}(\mathcal{X}) \times \mathcal{C}(\mathcal{X}) \to \mathbb{R}$ to denote the Hilbert metric of continuous functions,

$$d_H(\mu,\nu) = \log \max_{x,x'\in\mathcal{X}} \frac{\mu(x)\nu(x')}{\mu(x')\nu(x)}.$$

Note that $d_H(\mu,\nu) = d_H(1/\mu, 1/\nu)$ if $\mu(x) > 0$ and $\nu(x) > 0$ $\forall x \in \mathcal{X}$ and hence $d_H(L_\alpha\mu, L_\alpha\nu) = d_H(A_\alpha\mu, A_\alpha\nu)$. Under the above notations, we introduce the following existing result.

**Lemma C.1** (Birkhoff-Hopf Theorem Lemmens and Nussbaum [2012], see Lemma B.4 in Luise et al. [2019]). *Let* $\lambda = \frac{\exp(M_c/\gamma)-1}{\exp(M_c/\gamma)+1}$ *and* $\alpha \in \mathcal{M}_1^+(\mathcal{X})$. *Then for every* $u,v \in \mathcal{C}(\mathcal{X})$, *such that* $u(x) > 0, v(x) > 0$ *for all* $x \in \mathcal{X}$, *we have*

$$d_H(L_\alpha u, L_\alpha v) \leq \lambda d_H(u,v).$$

Note that from the definition of $d_H$, one has

$$\| \log \mu - \log \nu \|_\infty \leq d_H(\mu, \nu) = \max_x [\log \mu(x) - \log \nu(x)] + \max_x [\log \nu(x) - \log \mu(x)]$$
$$\leq 2 \| \log \mu - \log \nu \|_\infty.$$

In the following, we derive upper bound for $d_H(\mu, \nu)$ and use such bound to analyze the Lipschitz continuity of the Sinkhorn potentials $f$ and $g$.

Construct $\tilde{v} := A_\alpha u'$. Using the triangle inequality (which holds since $v(x), v'(x), \tilde{v}(x) > 0$ for all $x \in \mathcal{X}$), we have

$$d_H(v, v') \leq d_H(v, \tilde{v}) + d_H(\tilde{v}, v') \leq \lambda d_H(u, u') + d_H(\tilde{v}, v'),$$

where the second inequality is due to Lemma C.1. Similarly, Construct $\tilde{u} := A_\beta v'$. Apply Lemma C.1 again to obtain

$$d_H(u, u') \leq d_H(u, \tilde{u}) + d_H(\tilde{u}, u') \leq \lambda d_H(v, v') + d_H(\tilde{u}, u').$$

Together, we obtain

$$d_H(v, v') \leq \lambda^2 d_H(v, v') + d_H(\tilde{v}, v') + \lambda d_H(\tilde{u}, u') \leq \lambda^2 d_H(v, v') + d_H(\tilde{v}, v') + d_H(\tilde{u}, u'),$$

which leads to

$$d_H(v, v') \leq \frac{1}{1 - \lambda^2} [d_H(\tilde{v}, v') + d_H(\tilde{u}, u')].$$

To bound $d_H(\tilde{v}, v')$ and similarly $d_H(\tilde{u}, u')$, observe the following:

$$d_H(v', \tilde{v}) = d_H(L_{\alpha'} u', L_\alpha u') \leq 2 \| \log L_{\alpha'} u' - \log L_\alpha u' \|_\infty$$
$$= 2 \max_{x \in \mathcal{X}} | \nabla \log(a_x) ([L_{\alpha'} u'](x) - [L_\alpha u'](x)) | = 2 \max_{x \in \mathcal{X}} \frac{1}{a_x} |[L_{\alpha'} u'](x) - [L_\alpha u'](x)|$$
$$\leq 2 \max \{ \| 1/L_{\alpha'} u' \|_\infty, \| 1/L_\alpha u' \|_\infty \} \| L_{\alpha'} u' - L_\alpha u' \|_\infty, \qquad (30)$$

where $a_x \in [[L_{\alpha'} u'](x), [L_\alpha u'](x)]]$ in the second line is from the mean value theorem. Further, in the inequality we use $\max \{ \| 1/L_\alpha u' \|_\infty, \| 1/L_\alpha u' \|_\infty \} = \max \{ \| A_{\alpha'} u' \|_\infty, \| A_\alpha u' \|_\infty \} \leq \exp(2M_c/\gamma)$. Consequently, all we need to bound is the last term $\| L_{\alpha'} u' - L_\alpha u' \|_\infty$.

**Result (i)** We first note that $\forall x \in \mathcal{X}, \| l(x, \cdot) u'(\cdot) \|_{bl} < \infty$: In terms of $\| \cdot \|_\infty$

$$\| l(x, \cdot) u'(\cdot) \|_\infty \leq \| l(x, \cdot) \|_\infty \| u' \|_\infty \leq \exp(3M_c/\gamma) < \infty.$$

In terms of $\| \cdot \|_{lip}$, we bound

$$\| l(x, \cdot) u'(\cdot) \|_{lip} \leq \| l(x, \cdot) \|_\infty \| u' \|_{lip} + \| l(x, \cdot) \|_{lip} \| u' \|_\infty$$
$$\leq \exp(M_c/\gamma) G_c \exp(2M_c/\gamma)/\gamma + \exp(M_c/\gamma) G_c \exp(2M_c/\gamma)/\gamma$$
$$= 2G_c \exp(3M_c/\gamma)/\gamma < \infty.$$

Together we have $\| l(x, y) u'(y) \|_{bl} \leq \max \{ \exp(3M_c/\gamma), 2G_c \exp(3M_c/\gamma)/\gamma \}$. From the definition of the operator $L_\alpha$, we have

$$\| L_{\alpha'} u' - L_\alpha u' \|_\infty = \max_x | \int_\mathcal{X} l(x, y) u'(y) \mathbf{d}\alpha'(y) - \int_\mathcal{X} l(x, y) u'(y) \mathbf{d}\alpha(y) |$$
$$\leq \| l(x, y) u'(y) \|_{bl} d_{bl}(\alpha', \alpha).$$

All together we derive

$$d_H(v', v) \leq \frac{2 \exp(2M_c/\gamma) \| l(x, y) u'(y) \|_{bl}}{1 - \lambda^2} [d_{bl}(\alpha', \alpha) + d_{bl}(\beta', \beta)] \quad (\lambda = \frac{\exp(M_c/\gamma) - 1}{\exp(M_c/\gamma) + 1}).$$

Further, since $d_H(v', v) \geq \| \log v' - \log v \|_\infty = \frac{1}{\gamma} \| f' - f \|_\infty$, we have the result:

$$\| f' - f \|_\infty \leq \frac{2\gamma \exp(2M_c/\gamma) \| l(x, y) u'(y) \|_{bl}}{1 - \lambda^2} [d_{bl}(\alpha', \alpha) + d_{bl}(\beta', \beta)]. \qquad (31)$$

Similar argument can be made for $\| g' - g \|_\infty$.

**Result (ii)** Recall that $\alpha' = T_\phi \sharp \alpha$ and $\beta' = \beta$ with $T_\phi(x) = x + \phi(x)$. For simplicity we denote $f' = f_{T_\phi \sharp \alpha, \beta}$ and $g' = g_{T_\phi \sharp \alpha, \beta}$ and $f = f_{\alpha, \beta}$ and $g = g_{\alpha, \beta}$. We denote similarly $u'$, $v'$, $u$, and $v$. Use (30) and the change-of-variables formula of the push-forward measure to obtain

$$\|L_{T_\phi \sharp \alpha} u' - L_\alpha u'\|_\infty = \max_x \int [l(x, T_\phi(y)) u'(T_\phi(y)) - l(x, y) u'(y)] \mathbf{d}\alpha(y).$$

We now bound the integrand:

$$|l(x, T_\phi(y)) u'(T_\phi(y)) - l(x, y) u'(y)|$$
$$= |l(x, T_\phi(y)) u'(T_\phi(y)) - l(x, T_\phi(y)) u'(y)| + |l(x, T_\phi(y)) u'(y) - l(x, y) u'(y)|$$
$$\leq \exp(M_c/\gamma) \cdot \frac{G_c \exp(2M_c/\gamma)}{\gamma} \|\phi(y)\| + \exp(M_c/\gamma) \frac{G_c}{\gamma} \cdot \exp(2M_c/\gamma) \cdot \|\phi(y)\|$$
$$\leq \frac{2G_c \exp(3M_c/\gamma)}{\gamma} \cdot \|\phi(y)\|,$$

where we use the Lipschitz continuity of $u'$ for the first term and the Lipschitz continuity of $l$ for the second term.

## C.4 Proof of Lemma B.2

From the restriction on $d_{bl}(\alpha', \alpha) + d_{bl}(\beta', \beta)$ or the size of the mapping $\|\phi\|_\infty$, we always have $|f(x) + g(y) - f'(x) - g'(y)| < 1$ from Lemma B.1.
Denote the Sinkhorn potentials to $OT_\gamma(\alpha, \beta)$ and $OT_\gamma(\alpha', \beta')$ by $(f, g)$ and $(f', g')$ respectively. From the expression (14) of $\nabla f$ (and $\nabla f'$), we have

$$\|\nabla f(x) - \nabla f'(x)\| = \| \int_{\mathcal{X}} (h(x, y) - h'(x, y)) \nabla_x c(x, y) \mathbf{d}\beta(y) \|$$
$$= \| \int_{\mathcal{X}} h'(x, y) (\exp(f(x) + g(y) - f'(x) - g'(y)) - 1) \nabla_x c(x, y) \mathbf{d}\beta(y) \|$$
$$\leq \int_{\mathcal{X}} h'(x, y) |\exp(f(x) + g(y) - f'(x) - g'(y)) - 1| \|\nabla_x c(x, y)\| \mathbf{d}\beta(y)$$
$$\leq \int_{\mathcal{X}} 2h'(x, y) |f(x) + g(y) - f'(x) - g'(y)| \|\nabla_x c(x, y)\| \mathbf{d}\beta(y),$$

where $h'(x, y) := \exp(\frac{1}{\gamma}(f'(x) + g'(y) - c(x, y)))$, the second inequality holds since $|exp(x) - 1| < 2|x|$ when $|x| \leq 1$ and $|f(x) + g(y) - f'(x) - g'(y)| < 1$. We can use results from Lemma B.1 to bound the term $|f(x) + g(y) - f'(x) - g'(y)|$.

**Result (i):** Using (i) of Lemma B.1, we bound

$$\|\nabla f(x) - \nabla f'(x)\| \leq 2G_c G_{bl} [d_{bl}(\alpha', \alpha) + d_{bl}(\beta', \beta)].$$

**Result (ii):** Using (ii) of Lemma B.1, we bound

$$\|\nabla f(x) - \nabla f'(x)\| \leq 2G_c G_T \|\phi\|_\infty.$$

## C.5 Proof of Proposition 3.1

We will compute $DF_1[0]$ based on the definition of the Fréchet derivatives in Definition 1.1. The computation of $DF_2[0]$ follows similarly.
Denote $T_\psi = \mathcal{I} + \psi$. Note that we are interested in the case when $\psi = 0$ and hence $T_{\psi + \epsilon\phi}(x) = T_{\epsilon\phi}(x) = x + \epsilon\phi(x)$. Additionally, $T_\psi$ is the identity operator when $\psi = 0$ and hence $F_1[0] = OT_\gamma(\alpha, \beta)$. For simplicity, we drop the subscript of $T_{\epsilon\phi}$ ($\psi = 0$) and simply denote it by $T$ in the rest of the proof. Let $f$ and $g$ be the Sinkhorn potentials to $OT_\gamma(\alpha, \beta)$, by (5) and the optimality of $f$ and $g$, one has

$$OT_\gamma(\alpha, \beta) = \langle f, \alpha \rangle + \langle g, \beta \rangle.$$

However, $f$ and $g$ are not necessarily the optimal dual variables for $OT_\gamma(T_\sharp \alpha, \beta)$, so one has

$$OT_\gamma(T_\sharp \alpha, \beta) \geq \langle f, T_\sharp \alpha \rangle + \langle g, \beta \rangle - \gamma \langle h - 1, T_\sharp \alpha \otimes \beta \rangle.$$

Using the optimality from Lemma A.1, we have $\int_{\mathcal{X}} h(x, y)\mathbf{d}\beta(y) = 1$ and hence $\langle h-1, T_\sharp \alpha \otimes \beta \rangle = 0$. Subtracting the 1st equality from the last inequality,

$$\text{OT}_\gamma(T_\sharp \alpha, \beta) - \text{OT}_\gamma(\alpha, \beta) \geq \langle f, T_\sharp \alpha - \alpha \rangle.$$

Use the change-of-variables formula of the push-forward measure to obtain

$$\frac{1}{\epsilon}\langle f, T_\sharp \alpha - \alpha \rangle = \frac{1}{\epsilon}\int_{\mathcal{X}} \big((f \circ T)(x) - f(x)\big)\mathbf{d}\alpha(x) = \int_{\mathcal{X}} \nabla f(x + \epsilon'\phi(x))\phi(x)\mathbf{d}\alpha(x),$$

where $\epsilon' \in [0, \epsilon]$ is from the mean value theorem. Further use the Lipschitz continuity of $\nabla f$ in Lemma A.4, we have

$$\lim_{\epsilon \to 0} \frac{1}{\epsilon}\langle f, T_\sharp \alpha - \alpha \rangle = \int_{\mathcal{X}} \nabla f(x)\phi(x)\mathbf{d}\alpha(x).$$

Since $\phi \in \mathcal{H}^d$, we have $\phi(x) = \langle \phi, k(x, \cdot) \rangle_{\mathcal{H}^d}$ and hence

$$\lim_{\epsilon \to 0} \frac{1}{\epsilon}\big(\text{OT}_\gamma(T_\sharp \alpha, \beta) - \text{OT}_\gamma(\alpha, \beta)\big) \geq \langle \int \nabla f(x)k(x, \cdot)\mathbf{d}\alpha(x), \phi \rangle_{\mathcal{H}^d}.$$

Similarly, let $f'$ and $g'$ be the Sinkhorn potentials to $\text{OT}_\gamma(T_\sharp \alpha, \beta)$, using $f' \to f$ as $\epsilon \to 0$, we can have an upper bound

$$\lim_{\epsilon \to 0} \frac{1}{\epsilon}\big(\text{OT}_\gamma(T_\sharp \alpha, \beta) - \text{OT}_\gamma(\alpha, \beta)\big) \leq \langle \int_{\mathcal{X}} \lim_{\epsilon \to 0} \nabla f'(x + \epsilon'\phi(x))k(x, \cdot)\mathbf{d}\alpha(x), \phi \rangle_{\mathcal{H}^d}.$$

Since $\phi \in \mathcal{H}^d$, we have $\|\phi\|_{2,\infty} \leq M_{\mathcal{H}}\|\phi\|_{\mathcal{H}^d} < \infty$ with $M_{\mathcal{H}} \in \mathbb{R}_+$ being a constant. Using Lemma B.1, we have that $\nabla f'$ is Lipschitz continuous with respect to the mapping

$$\lim_{\epsilon \to 0} \|\nabla f'(x + \epsilon'\phi(x)) - \nabla f(x + \epsilon'\phi(x))\| \leq \lim_{\epsilon \to 0} \epsilon G_T \|\phi\|_{2,\infty} = 0.$$

Besides, using Lemma A.4 we have that $\nabla f$ is continuous and hence $\lim_{\epsilon \to 0} \nabla f(x + \epsilon'\phi(x)) = \nabla f(x)$. Consequently we have $\lim_{\epsilon \to 0} \nabla f'(x + \epsilon'\phi(x)) = \nabla f(x)$ and hence

$$\lim_{\epsilon \to 0} \frac{1}{\epsilon}\big(\text{OT}_\gamma(T_\sharp \alpha, \beta) - \text{OT}_\gamma(\alpha, \beta)\big) = \langle \nabla f(x)k(x, \cdot)\mathbf{d}\alpha(x), \phi \rangle_{\mathcal{H}^d}.$$

From Definition 1.1, we have the result of $DF_1[0]$. The result of $DF_2[0]$ can be obtained similarly.

## C.6 Proof of Lemma 4.2

From Proposition 3.1 and (13), we recall the expression of $D\mathcal{S}_\alpha[0]$ by

$$D\mathcal{S}_\alpha[0] = \int_{\mathcal{X}} [\frac{1}{n}\sum_{i=1}^n \nabla f_{\alpha,\beta_i}(x) - \nabla f_{\alpha,\alpha}(x)]k(x, y)\mathbf{d}\alpha(x), \tag{32}$$

and we have $\mathcal{T}[\alpha](x) = x - \eta D\mathcal{S}_\alpha[0](x)$. Consequently, using Corollary B.1 we have

$$\begin{aligned}\|\mathcal{T}[\alpha]\|_{lip} &= \max_{x \neq y} \frac{\|\mathcal{T}[\alpha](x) - \mathcal{T}[\alpha](y)\|}{\|x - y\|} = \max_{x \neq y} \frac{\|x - y - \eta(D\mathcal{S}_\alpha[0](x) - D\mathcal{S}_\alpha[0](y))\|}{\|x - y\|} \\ &\leq 1 + \eta\|D\mathcal{S}_\alpha[0]\|_{lip} \leq 1 + \eta G_c G_k.\end{aligned}$$

The following lemma states that $\mathcal{T}[\alpha]$ is Lipschitz w.r.t. $\alpha$ in terms of the bounded Lipschitz norm.

**Lemma C.2.** *For any* $y \in \mathcal{X}$ *and any* $\alpha, \alpha' \in \mathcal{M}_1^+(\mathcal{X})$, *we have*

$$\|\mathcal{T}[\alpha](y) - \mathcal{T}[\alpha'](y)\|_{2,\infty} \leq \eta \max\{dL_f D_k + dG_c G_k, D_k L_{bl}\}d_{bl}(\alpha', \alpha).$$

We defer the proof to Appendix C.6.1. Based on such lemma, for any $h$ with $\|h\|_{bl} \leq 1$, we have

$$\begin{aligned}|\langle h, \mathcal{T}[\alpha]_\sharp \alpha \rangle - \langle h, \mathcal{T}[\alpha']_\sharp \alpha' \rangle| &= |\langle h \circ \mathcal{T}[\alpha], \alpha \rangle - \langle h \circ \mathcal{T}[\alpha'], \alpha' \rangle| \\ &\leq |\langle h \circ \mathcal{T}[\alpha], \alpha \rangle - \langle h \circ \mathcal{T}[\alpha], \alpha' \rangle| + |\langle h \circ \mathcal{T}[\alpha], \alpha' \rangle - \langle h \circ \mathcal{T}[\alpha'], \alpha' \rangle|.\end{aligned}$$

We now bound these two terms individually: For the first term,

$$\begin{aligned}|\langle h \circ \mathcal{T}[\alpha], \alpha \rangle - \langle h \circ \mathcal{T}[\alpha], \alpha' \rangle| &\leq \|h \circ \mathcal{T}[\alpha]\|_{bl}d_{bl}(\alpha, \alpha') \\ &\leq \max\{\|h\|_\infty, \|h\|_{lip}\|\mathcal{T}[\alpha]\|_{lip}\}d_{bl}(\alpha, \alpha') \leq (1 + \eta G_c G_k)d_{bl}(\alpha, \alpha');\end{aligned}$$

And for the second term, use Lemma C.2 to derive

$$|\langle h \circ \mathcal{T}[\alpha], \alpha' \rangle - \langle h \circ \mathcal{T}[\alpha'], \alpha' \rangle|$$
$$\leq \|h \circ \mathcal{T}[\alpha] - h \circ \mathcal{T}[\alpha']\|_\infty \leq \|h\|_{lip} \max_{x \in \mathcal{X}} \|\mathcal{T}[\alpha](x) - \mathcal{T}[\alpha'](x)\|$$
$$\leq \eta \max\{dL_f D_k + dG_c G_k, D_k L_{bl}\} d_{bl}(\alpha', \alpha).$$

Combining the above inequalities, we have the result

$$d_{bl}(\mathcal{T}[\alpha]_\sharp \alpha, \mathcal{T}[\alpha']_\sharp \alpha') \leq (1 + \eta G_c G_k + \eta \max\{dL_f D_k + dG_c G_k, D_k L_{bl}\}) d_{bl}(\alpha', \alpha).$$

### C.6.1 Proof of Lemma C.2

Recall the definition of $\mathcal{T}[\alpha](x) = x - \eta D\mathcal{S}_\alpha[0](x)$, where the functional $\mathcal{S}_\alpha$ is defined in (9) and the Fréchet derivative is computed in (13). Denote $\xi(x) := \frac{1}{n} \sum_{i=1}^n \nabla f_{\alpha,\beta_i}(x) - \nabla f_{\alpha,\alpha}(x)$ and $\xi'(x) := \frac{1}{n} \sum_{i=1}^n \nabla f_{\alpha',\beta_i}(x) - \nabla f_{\alpha',\alpha'}(x)$. For any $y \in \mathcal{X}$, we have

$$\|\mathcal{T}[\alpha](y) - \mathcal{T}[\alpha'](y)\| \leq \eta \|D\mathcal{S}_\alpha[0](y) - D\mathcal{S}_{\alpha'}[0](y)\|$$
$$\leq \eta \| \int_{\mathcal{X}} \xi(x) k(x,y) \mathbf{d}\alpha(x) - \int_{\mathcal{X}} \xi'(x) k(x,y) \mathbf{d}\alpha'(x) \|$$
$$\leq \eta \| \int_{\mathcal{X}} (\xi(x) - \xi'(x)) k(x,y) \mathbf{d}\alpha(x) \| + \eta \| \int_{\mathcal{X}} \xi'(x) k(x,y) \mathbf{d}(\alpha(x) - \alpha'(x)) \|.$$

For the first term, use Lemma B.2 to bound

$$\| \int_{\mathcal{X}} (\xi(x) - \xi'(x)) k(x,y) \mathbf{d}\alpha(x) \|$$
$$= \| \int_{\mathcal{X}} \left( \frac{1}{n} [\sum_{i=1}^n \nabla f_{\alpha,\beta_i}(x) - \nabla f_{\alpha',\beta_i}(x)] - \nabla f_{\alpha,\alpha}(x) + \nabla f_{\alpha',\alpha'}(x) \right) k(x,y) \mathbf{d}\alpha(x) \|$$
$$\leq D_k L_{bl} d_{bl}(\alpha', \alpha).$$

For the second term, we bound

$$\| \int_{\mathcal{X}} \xi'(x) k(x,y) \mathbf{d}(\alpha(x) - \alpha'(x)) \| \leq \| \int_{\mathcal{X}} \xi'(x) k(x,y) \mathbf{d}(\alpha(x) - \alpha'(x)) \|_1$$
$$\leq \sum_{i=1}^d | \int_{\mathcal{X}} [\xi'(x)]_i k(x,y) \mathbf{d}(\alpha(x) - \alpha'(x)) |$$
$$\leq \sum_{i=1}^d \|[\xi'(x)]_i k(\cdot, y)\|_{bl} d_{bl}(\alpha', \alpha).$$

Therefore, we only need to bound $\sum_{i=1}^d \|[\frac{1}{n} \sum_{i=1}^n \nabla f_{\alpha',\beta_i}(x) - \nabla f_{\alpha',\alpha'}(x)]_i k(x,y)\|_{bl}$. In terms of $L^\infty$ norm, we have

$$\sum_{i=1}^d \|[\frac{1}{n} \sum_{i=1}^n \nabla f_{\alpha',\beta_i}(\cdot) - \nabla f_{\alpha',\alpha'}(\cdot)]_i k(\cdot, y)\|_\infty \leq dD_k \|[\nabla f_{\alpha',\beta_i}]_i\|_\infty \leq dD_k G_c.$$

In terms of $\| \cdot \|_{lip}$, denote $\tilde{\nabla}(x) = \frac{1}{n} \sum_{i=1}^n \nabla f_{\alpha',\beta_i}(x) - \nabla f_{\alpha',\alpha'}(x)$. For all $x, x' \in \mathcal{X}$, we have

$$\frac{|[\tilde{\nabla}(x)]_i k(x,y) - [\tilde{\nabla}(x')]_i k(x',y)|}{\|x - x'\|}$$
$$\leq \frac{|[\tilde{\nabla}(x)]_i k(x,y) - [\tilde{\nabla}(x')]_i k(x,y)| + |[\tilde{\nabla}(x')]_i k(x,y) - [\tilde{\nabla}(x')]_i k(x',y)|}{\|x - x'\|}$$
$$\leq L_f D_k + G_c G_k,$$

and hence $\sum_{i=1}^d \|[\frac{1}{n} \sum_{i=1}^n \nabla f_{\alpha',\beta_i}(\cdot) - \nabla f_{\alpha',\alpha'}(\cdot)]_i k(\cdot, y)\|_{lip} \leq dL_f D_k + dG_c G_k$. All together, we have for any $y \in \mathcal{X}$

$$\|\mathcal{T}[\alpha](y) - \mathcal{T}[\alpha'](y)\| \leq \eta \max\{dL_f D_k + dG_c G_k, D_k L_{bl}\} d_{bl}(\alpha', \alpha).$$

## C.7 Proof of Lemma 4.1

We first recall a proposition from Feydy et al. [2019], which shows that the dual potentials (more precisely, their extensions to the whole domain) are the variations of $\mathrm{OT}_\gamma$ w.r.t. the underlying probability measure.

**Definition C.1.** *We say $h \in \mathcal{C}(\mathcal{X})$ is the first-order variation of a functional $F : \mathcal{M}_1^+(\mathcal{X}) \to \mathbb{R}$ at $\alpha \in \mathcal{M}_1^+(\mathcal{X})$ if for any displacement $\xi = \beta - \alpha$ with $\beta \in \mathcal{M}_1^+(\mathcal{X})$, we have*
$$F(\alpha + t\xi) = F(\alpha) + t\langle h, \xi\rangle + o(t).$$
*Further we denote $h = \nabla_\alpha F(\alpha)$.*

**Lemma C.3.** *The first-order variation of $\mathrm{OT}_\gamma(\alpha, \beta)(\alpha \neq \beta)$ with respect to the measures $\alpha$ and $\beta$ is the corresponding Sinkhorn potential, i.e. $\nabla_{(\alpha,\beta)}\mathrm{OT}_\gamma(\alpha, \beta) = (f_{\alpha,\beta}, g_{\alpha,\beta})$. Further, if $\alpha = \beta$, we have $\nabla_\alpha \mathrm{OT}_\gamma(\alpha, \alpha) = 2f_{\alpha,\alpha}$.*

Recall that $\alpha^{t+1} = \mathcal{T}[\alpha^t]_\sharp \alpha^t$ where the push-forward mapping is of the form $\mathcal{T}[\alpha^t](x) = x - \eta D\mathcal{S}_{\alpha^t}[0](x)$ with $D\mathcal{S}_{\alpha^t}[0]$ given in (13). Using the convexity of $\mathcal{S}_\gamma$ and Lemma C.3, we have

$$\mathcal{S}_\gamma(\alpha^{t+1}) - \mathcal{S}_\gamma(\alpha^t)$$

$$\leq \langle \nabla_\alpha \mathcal{S}_\gamma(\alpha)|_{\alpha=\alpha^{t+1}}, \alpha^{t+1} - \alpha^t \rangle \qquad\qquad \text{\# convexity of } \mathcal{S}_\gamma$$

$$= \langle \frac{1}{n}\sum_{i=1}^n f_{\alpha^{t+1},\beta_i} - f_{\alpha^{t+1},\alpha^{t+1}}, \mathcal{T}[\alpha^t]_\sharp \alpha^t - \alpha^t\rangle \qquad\qquad \text{\# Lemma C.3}$$

$$= \langle [\frac{1}{n}\sum_{i=1}^n f_{\alpha^{t+1},\beta_i} - f_{\alpha^{t+1},\alpha^{t+1}}] \circ \mathcal{T}[\alpha^t] - [\frac{1}{n}\sum_{i=1}^n f_{\alpha^{t+1},\beta_i} - f_{\alpha^{t+1},\alpha^{t+1}}], \alpha^t\rangle. \quad \text{\# change-of-variables}$$

For succinctness, denote $\xi^t := \frac{1}{n}\sum_{i=1}^n f_{\alpha^t,\beta_i} - f_{\alpha^t,\alpha^t}$. Hence, we have

$$\mathcal{S}_\gamma(\alpha^{t+1}) - \mathcal{S}_\gamma(\alpha^t) \leq \langle \xi^{t+1}\circ\mathcal{T}[\alpha^t] - \xi^{t+1}, \alpha^t\rangle = \int \xi^{t+1}(x - \eta D\mathcal{S}_{\alpha^t}[0](x)) - \xi^{t+1}(x)\mathbf{d}\alpha^t(x)$$

$$= -\eta\int \langle\nabla\xi^{t+1}(x - \eta' D\mathcal{S}_{\alpha^t}[0](x)), D\mathcal{S}_{\alpha^t}[0](x)\rangle \mathbf{d}\alpha^t(x),$$

where the last equality is from the mean value theorem with $\eta' \in [0, \eta]$. We now bound the integral by splitting it into three terms and analyze them one by one.

$$\int_\mathcal{X} \langle\nabla\xi^{t+1}(x - \eta' D\mathcal{S}_{\alpha^t}[0](x)), D\mathcal{S}_{\alpha^t}[0](x)\rangle \mathbf{d}\alpha^t(x)$$

$$= \int_\mathcal{X} \langle\nabla\xi^t(x), D\mathcal{S}_{\alpha^t}[0](x)\rangle \mathbf{d}\alpha^t(x) \qquad\qquad\qquad\qquad\qquad ①$$

$$+ \int_\mathcal{X} \langle\nabla\xi^t(x - \eta' D\mathcal{S}_{\alpha^t}[0](x)) - \nabla\xi^t(x), D\mathcal{S}_{\alpha^t}[0](x)\rangle \mathbf{d}\alpha^t(x) \qquad ②$$

$$+ \int_\mathcal{X} \langle\nabla\xi^{t+1}(x - \eta' D\mathcal{S}_{\alpha^t}[0](x)) - \nabla\xi^t(x - \eta' D\mathcal{S}_{\alpha^t}[0](x)), D\mathcal{S}_{\alpha^t}[0](x)\rangle \mathbf{d}\alpha^t(x). \quad ③$$

For ①, since $D\mathcal{S}_{\alpha^t}[0] \in \mathcal{H}^d$, we have $D\mathcal{S}_{\alpha^t}[0](x) = \langle D\mathcal{S}_{\alpha^t}[0], k(x, \cdot)\rangle$ and hence

$$\int_\mathcal{X} \langle\nabla\xi^t(x), D\mathcal{S}_{\alpha^t}[0](x)\rangle \mathbf{d}\alpha^t(x) = \int \langle\nabla\xi^t(x)k(x, \cdot), D\mathcal{S}_{\alpha^t}[0]\rangle_{\mathcal{H}^d} \mathbf{d}\alpha^t(x)$$

$$= \|D\mathcal{S}_{\alpha^t}[0]\|_{\mathcal{H}^d}^2 = \mathbf{S}(\alpha^t, \{\beta_i\}_{i=1}^n),$$

where the last equality is from the Definition 4.1 and the expression of $D\mathcal{S}_\alpha[0]$ in (13).

For ②, note that the summands of $\nabla\xi^t$ are of the form $\nabla f_{\alpha,\beta}$ (or $\nabla f_{\alpha,\alpha}$) which is proved to be Lipschitz in Lemma A.4. Consequently, we bound

$$|\int \langle\nabla\xi^t(x - \eta' D\mathcal{S}_{\alpha^t}[0](x)) - \nabla\xi^t(x), D\mathcal{S}_{\alpha^t}[0](x)\rangle \mathbf{d}\alpha^t(x)|$$

$$\leq \int \|\nabla\xi^t(x - \eta' D\mathcal{S}_{\alpha^t}[0](x)) - \nabla\xi^t(x)\| \|D\mathcal{S}_{\alpha^t}[0](x)\| \mathbf{d}\alpha^t(x)$$

$$\leq \int 2L_f\eta \|D\mathcal{S}_{\alpha^t}[0](x)\|^2 \mathbf{d}\alpha^t(x) \qquad\qquad\qquad \text{\# Lemma A.4}$$

$$\leq 2\eta L_f M_\mathcal{H}^2 \|D\mathcal{S}_{\alpha^t}[0]\|_{\mathcal{H}^d}^2 = 2\eta L_f M_\mathcal{H}^2 \mathbf{S}(\alpha^t, \{\beta_i\}_{i=1}^n). \qquad \text{\# see (1)}$$

where we use $\forall f \in \mathcal{H}^d, \exists M_{\mathcal{H}} > 0$ s.t. $\|f(x)\| \leq M_{\mathcal{H}}\|f\|_{\mathcal{H}^d}, \forall x \in \mathcal{X}$ in the third inequality.

For ③, similar to ②, the summands of $\nabla \xi^t$ are proved to be Lipschitz in (ii) of Lemma B.2, and hence we bound

$$|\int \langle \nabla \xi^{t+1}(x - \eta' D\mathcal{S}_{\alpha^t}[0](x)) - \nabla \xi^t(x - \eta' D\mathcal{S}_{\alpha^t}[0](x)), D\mathcal{S}_{\alpha^t}[0](x)\rangle \mathbf{d}\alpha^t(x)|$$

$$\leq \int \|\nabla \xi^{t+1}(x - \eta' D\mathcal{S}_{\alpha^t}[0](x)) - \nabla \xi^t(x - \eta' D\mathcal{S}_{\alpha^t}[0](x))\| \|D\mathcal{S}_{\alpha^t}[0](x)\| \mathbf{d}\alpha^t(x)$$

$$\leq \int \sqrt{d}\eta L_T \|D\mathcal{S}_{\alpha^t}[0]\|_{2,\infty} \|D\mathcal{S}_{\alpha^t}[0](x)\| \mathbf{d}\alpha^t(x) \qquad \qquad \text{\# Lemma B.2}$$

$$\leq 2\eta\sqrt{d}L_T M_{\mathcal{H}}^2 \|D\mathcal{S}_{\alpha^t}[0]\|_{\mathcal{H}^d}^2 = 2\eta\sqrt{d}L_T M_{\mathcal{H}}^2 \mathbf{S}(\alpha^t, \{\beta_i\}_{i=1}^n) \qquad \qquad \text{\# see (1)}$$

Combining the bounds on ①, ②, ③, we have:

$$\mathcal{S}_\gamma(\alpha^{t+1}) - \mathcal{S}_\gamma(\alpha^t) \leq -\eta(1 - 2\eta L_f M_{\mathcal{H}}^2 - 2\eta\sqrt{d}L_T M_{\mathcal{H}}^2)\mathbf{S}(\alpha^t, \{\beta_i\}_{i=1}^n),$$

which leads to the result when we set $\eta \leq \min\{\frac{1}{8L_f M_{\mathcal{H}}^2}, \frac{1}{8\sqrt{d}L_T M_{\mathcal{H}}^2}\}$.

# D    A Discussion on the Global Optimality

## D.1    Proof of Theorem 4.3

We first show $\int_{\mathcal{X}} \|\xi(x)\|^2 \mathbf{d}\alpha(x) < \infty$:

$$\int_{\mathcal{X}} \|\xi(x)\|^2 \mathbf{d}\alpha(x) = \int_{\mathcal{X}} \|\frac{1}{n}\sum_{i=1}^{n} \nabla f_{\alpha,\beta_i}(x) - \nabla f_{\alpha,\alpha}(x)\|_2^2 \mathbf{d}\alpha(x)$$

$$= \int_{\mathcal{X}} 2\|\frac{1}{n}\sum_{i=1}^{n} \nabla f_{\alpha,\beta_i}(x)\|^2 + 2\|\nabla f_{\alpha,\alpha}(x)\|_2^2 \mathbf{d}\alpha(x) \le 4G_f < \infty$$

(i) $\mathbf{S}(\alpha, \{\beta_i\}_{i=1}^{n}) = 0$ & $\mathrm{supp}(\alpha) = \mathcal{X} \Rightarrow \max_{\beta \in \mathcal{M}_1^+(\mathcal{X})} \langle -\nabla_\alpha \mathcal{S}_\gamma(\alpha), \beta - \alpha \rangle \le 0$:
From the integrally strictly positive definiteness of the kernel function $k(x, x')$, we have that $\int_{\mathcal{X}} \|\xi(x)\|^2 \mathbf{d}\alpha(x) = 0$ which implies $\nabla \xi = \frac{1}{n}\sum_{i=1}^{n} \nabla f_{\alpha,\beta_i} - \nabla f_{\alpha,\alpha}(x) = 0$ for all $x \in \mathrm{supp}(\alpha)$. Further, we have that $\xi$ is a constant function on $\mathcal{X}$ by $\mathrm{supp}(\alpha) = \mathcal{X}$. Since we can shift the Sinkhorn potential by a constant amount without losing its optimality, we can always ensure that $\xi$ is exactly a zero function. This implies the optimality condition of the Sinkhorn barycenter problem: $\max_{\beta \in \mathcal{M}_1^+(\mathcal{X})} \langle -\nabla_\alpha \mathcal{S}_\gamma(\alpha), \beta - \alpha \rangle \le 0$.
(ii) Using Theorem 4.1 and (i), one directly has the result.

## D.2    Fully Supported Property of SD at Finite Time

WLOG, suppose that $c(x, y) = \infty$ if $x \notin \mathcal{X}$. From the monotonicity of Lemma 4.1, the support of $\alpha^t$ will not grow beyond $\mathcal{X}$. Let $p^t$ be the density function of $\alpha^t$. The density $p^{t+1}$ is given by $p^{t+1}(x) = p^t(\mathcal{T}[\alpha^t]^{-1}(x)) |\det(\nabla \mathcal{T}[\alpha^t]^{-1}(x))|$, where $\mathcal{T}[\alpha^t]$ is the mapping defined in (11). For a sufficiently small step size, the determinant is always positive. Consequently, $p^{t+1}(x) = 0$ implies $p^t(\mathcal{T}[\alpha^t]^{-1}(x)) = 0$ which is impossible since $p^t$ is f.s. Therefore, $p^{t+1}$ is also a.c. and f.s.

## D.3    Review the Assumptions for Global Convergence in Previous Works

We briefly describe the assumptions required by previous works Arbel et al. [2019], Mroueh et al. [2019] to guarantee the global convergence to the MMD minimization problem. We emphasize that both of these works make assumptions on the ENTIRE measure sequence. In the following, we use $\nu_p$ to denote the target measure.

In Mroueh et al. [2019], given a measure $\nu \in \mathcal{M}_1^+(\mathcal{X})$, Mroueh et al. [2019] define the Kernel Derivative Gramian Embedding (KDGE) of $\nu$ by

$$D(\nu) := \mathbb{E}_{x \sim \nu} \left([J\Phi(x)]^\top J\Phi(x)\right), \tag{33}$$

where $\Phi$ is the feature map of a given RKHS and $J\Phi$ denotes its Jacobian matrix. Further denote the classic Kernel Mean Embedding (KME) by

$$\boldsymbol{\mu}(\nu) := \mathbb{E}_{x \sim \nu} \Phi(x). \tag{34}$$

SoD requires the entire variable measure sequence $\{\nu_q\}, q \ge 0$ to satisfy for any measure $\nu_q$ such that $\delta_{p,q} := \boldsymbol{\mu}(\nu_q) - \boldsymbol{\mu}(\nu_p) \ne 0$

$$D(\nu)\delta_{p,q} \ne 0. \tag{35}$$

In Arbel et al. [2019], Arbel et al. [2019] proposed two types of assumptions such that either of them leads to the global convergence of their (noisy) gradient flow algorithm. Specifically, denote the squared weighted Sobolev semi-norm of a function $f$ in an RKHS with respect to a measure $\nu$ by $\|f\|_{\dot{H}(\nu)} = \int_{\mathcal{X}} \|\nabla f(x)\|^2 d\nu(x)$. Given two probability measures on $\mathcal{X}$, $\nu_p$ and $\nu_q$, define the weighted negative Sobolev distance $\|\nu_p - \nu_q\|_{\dot{H}(\nu)^{-1}(\nu)}$ by

$$\|\nu_p - \nu_q\|_{\dot{H}(\nu)^{-1}(\nu)} = \sup_{f \in L_2(\nu), \|f\|_{\dot{H}(\nu)} \le 1} |\int_{\mathcal{X}} f(x)\nu_p(x) - \int_{\mathcal{X}} f(x)\nu_q(x)|. \tag{36}$$

In Proposition 7 of Arbel et al. [2019], if for the entire variable measure sequence $\{\nu_q\}$ generated by their gradient flow algorithm, $\|\nu_p - \nu_q\|_{\dot{H}(\nu)^{-1}(\nu)}$ is always bounded, then $\nu_q$ weakly converges to

$\nu_p$ under the MMD sense.

Further, the authors also propose another noisy gradient flow algorithm and provide its global convergence guarantee under a different assumption: Let $f_{\nu_p, \nu_q}$ be the unnormalized witness function to $\mathrm{MMD}(\nu_p, \nu_q)$. Let $\mu$ be the standard gaussian distribution and let $\beta > 0$ be a noise level. Denote $\mathcal{D}_\beta(\nu_q) := \mathbb{E}_{x \sim \nu_q, \mu}[\|\nabla f_{\nu_p, \nu_q}(x + \beta\mu)\|^2]$. The noisy gradient flow algorithm globally converges if for all $n$ there exists a noise level $\beta_n$ such that

$$8\lambda^2 \beta_n^2 \mathrm{MMD}(\nu_p, \nu_n) \leq \mathcal{D}_{\beta_n}(\nu_n), \tag{37}$$

and $\sum_{i=0}^{n} \beta_i^2 \to \infty$. Here $\lambda$ is some problem dependent constant.

# E   Implementation

The code to reproducing the experimental results can be found in the following link: `https://github.com/shenzebang/Sinkhorn_Descent`. Our implementation is based on Pytorch and geomloss[4].

## Footnotes

[4]`https://www.kernel-operations.io/geomloss/`