[Reviews · NeurIPS 2020]

Review 1

Summary and Contributions: The paper studies the problem of computing the barycenter of a set of probability distributions under the Sinkhorn divergence. Unlike previous approaches which directly consider an optimization problem on the space of probability measures, this work recasts the Sinkhorn barycenter problem as an instance of unconstrained functional optimization. The main contributions are the following: 1) a new method called Sinkhorn Descent (SD) for solving Sinkhorn barycenter problem based on finding a pushforward map in a RKHS that gives the direction of steepest descent . 2) a proof that SD preserves weak convergence of empirical measures 3) Introducing the notion of Kernelized Sinkhorn Barycenter Discrepancy (KSBD), they provide a non-asymptotic analysis of the algorithm. They showed that under some assumptions KSBD is a valid discrepancy to characterize the optimality of the solution.

Strengths: I think the paper is clear and very well written. Interpreting the optimization problem on measures as a problem on functions is an interesting and natural idea that was already applied, in a similar fashion, in Mroueh et al. “Sobolev Descent” for instance. Results are theoretically grounded and well explained and the authors provide a complete analysis of the method, which is novel in the framework of Sinkhorn barycenters and provides a good contribution to the literature.

Weaknesses: While I think the paper makes a good contribution, there are some limitation at the present stage: - [Remark 3.1] While it has been done in previous works, I think that a deeper understanding of those cases where modelling the pushforward P in (8) as a composition of perturbation in an RKHS does not introduce an error, would increase the quality of the work. Alternatively, trying to undestand the kind of error that this parametrization introduces would be valuable too. - The analysis does not cover explicitly what happens when the input measures \beta_i are absolutely continuous and one has to rely on samples. How does the sampling part impact the bound? - The experiments are limited to toy data. There is a range of problems with real data where barycenters can be used and it would be interesting to show performance of the method in those settings too.

Correctness: To the best of my knowledge, the results and method proposed in the paper are the correct.

Clarity: I find the paper is well written and easy to follow. I would suggest the author put the definition of d_bl (in eq(19) ) in the main text (and not only in the appendix)

Relation to Prior Work: It is clearly discussed how the work differs from previous contributions, both in terms of existing methods to compute Sinkhorn barycenters and in terms of other papers using functional gradient descent for different goals.

Reproducibility: Yes

Additional Feedback: I would like the author to clarify the following, if possible and to comment on what I pointed out as weaknesses in the section above. 1. As stated in lines 148-152, it is possible to cast the minimization problem in terms of maps rather than measures providing that the initial alpha_0 is regular (so that a pushforward always exist). However, the algorithm starts with alpha_0 equal to a finite set of particles. Does this introduce a limitation with respect to what can be capture in practise and how ? Will you start from a sufficiently large number of particles? 2. This is a minor detail: line 637 of the Appendix and subsequent few lines say that the gradients of OT_gamma(alpha,beta) are the potentials. I believe it would be more precise to mention that the gradients correspond to the *extensions* of the potentials on the whole domain. Typos (not exhaustive list): - line 659 in the brakets -> \nabla f_{alpha,alpha} - a few lines in the proofs of supplementary are too long for the page ########### AFTER REBUTTAL ########### Thanks for posting the feedback. After reading it together with the other reviews, I am leaving the score unchanged. I would suggest the authors include more information on the dependence on the number of particles of alpha_0, either theoretically or empirically.


Review 2

Summary and Contributions: They consider the problem of computing Sinkhorn barycenters (ie barycenter with de-biased entropic OT cost). They propose an algorithm Sinkhorn Descent based on functional GD. They show SD converges (but only to a stationary point and at a quite slow rate, scaling in exp(1/gamma)). Yet, the convergence can be improved to global minimizer under some technical assumptions (thm 4.3), and empirically convergence seems very fast for the presented runs.

Strengths: A great feature of SD is that it can be run in high dimensions. In comparison, previous algorithms are essentially limited to dimension 2 or 3. The empirical results show impressive convergence for the presented datasets, and the experiments well demonstrate the superiority of SD over the previous FW alg.

Weaknesses: A major drawback of the theoretical convergence is that per-iteration descent (Lem 4.1) scales as exp(1/gamma), where gamma is the regularization. Eg, for gamma=0.01, the decrease is ~exp(-100) which is basically 0. This issue is acknowledged in footnote 1, and it seems this is a hard problem to improve... How does this impact practice? I do not see gamma reported in the experimental section. How large is gamma in your experiments? Do you see worse empirical performance as gamma increases? In particular, is it exponentially bad in gamma as the theory suggests? Can you solve problems to reasonable accuracy with gamma = 10, 30, 100, 300? This small gamma means Sinkhorn cost is very far from OT. This is worth emphasizing in the paper. The experiments well demonstrate the superiority of SD over FW. What about other algs, eg Iterative Bregman Projections? Can you use logsumexp trick in SD as in standard Sinkhorn? This is important for numerical stability since exponentials are taken with large powers.

Correctness: Seems correct, but I did not read the supp materials.

Clarity: Yes. (typos: line 253 "ENTIRE", line 263 "the grid")

Relation to Prior Work: Yes.

Reproducibility: Yes

Additional Feedback: I am on the boundary because I am unsure about the Q above (and 1 below). I'd greatly appreciate clarification about these Q to help me understand/score. (Q: in Figs 1ac, is the y-axis the suboptimality or just the barycenter function value? If the former, then this seems to be very large error it converges to? If the latter, then please ignore this.) POST-REBUTTAL: Thanks for answering my questions. I adjust my score from 6 to 7 accordingly.


Review 3

Summary and Contributions: The article presents and study a new algorithm to compute Sinkhorn barycenter, a proxy of Wasserstein barycenter. Namely, the objective is to compute the measure with the least average squared Sinkhorn distance to a set of fixed measures. They prove a linear convergence in the objective of their algorithm as well as a mean-field convergence.

Strengths: The problem studied is pretty fashionable in these days, and is very relevant to the NeurIPS community. The convergences results are the first of their kind to obtain bounds with linear dependence on the dimension. Previous works had exponential dependence on the dimension. In particular, it allows them to compute the barycenter of Gaussian measures in dimension 100. The assumptions of their results seem pretty mild. Although I did not check the proofs in details, the results are pretty sound to me.

Weaknesses: The constants in the bounds depend linearly on the dimension, although they depends exponentially on the regularization parameter. If Sinkhorn distance is thought as a proxy of the Wasserstein distance, this seems to be a hidden dependance on the dimension, since the regularization parameter plays the role of an interpolation between MMD and Wasserstein distances, and MMD distances are more blind to the dimension. This is not discussed in the paper. The results also have an exponential dependence on an assumed uniform upper bound on the cost. For the classical quadratic cost, this imply an exponential dependence on the dimension for the case of measures supported on [0,1]^d for instance. This is not discussed either. Also, their algorithm uses as input the Sinkhon potential. They treat it as a blackbox that and use an algorithm from Genevay et al. 2016. I think a brief description of the numerical cost of this blackbox would be useful. They compoare their results with none of the classical tools for computing barycenter! For instance: Benamou, J. D., Carlier, G., Cuturi, M., Nenna, L., & Peyré, G. (2015). Iterative Bregman projections for regularized transportation problems Chewi, S., Maunu, T., Rigollet, P., & Stromme, A. J. (2020). Gradient descent algorithms for Bures-Wasserstein barycenters. Cuturi, M., & Doucet, A. (2014). Fast computation of Wasserstein barycenters. Claici, S., Chien, E., & Solomon, J. (2018). Stochastic wasserstein barycenters. Although they claim to compute a Sinkhorn barycenter instead of a Wasserstein barycenter, given the visual experiment they provide, the objective is the same. Also, they do not tell what regularization parameter they choose.

Correctness: The proofs seems correct to me. Although, the numerical experiment is missing major comparison to be really valuable.

Clarity: The paper is pretty well written in general. I would argue that they are missing references, such as the paper Agueh, M., & Carlier, G. (2011). Barycenters in the Wasserstein space. that initiated the whole work on Wasserstein barycenter.

Relation to Prior Work: The relation to prior work is well document although, as mentionned above, comparison for numerical experiments is weak, and discussion in the dependence on the dimension with previous work ought to be more thorough as well.

Reproducibility: Yes

Additional Feedback: ===== Post rebuttal ===== I am satisfied with the author's rebuttal and after reading other reviews, I keep my score. ====================


Review 4

Summary and Contributions: This work proposes a novel functional gradient descent algorithm for Sinkhorn barycenter problem, named Sinkhorn Descent. The proposed method converges to the global minimizer at a sublinear rate, and also preserved the weak convergence of empirical measures. The method scales linearly with respect to the problem dimension.

Strengths: The proposed method has nice convergence properties and good complexity. It does not require a prespecified support for the barycenter.

Weaknesses: It would be better if the empirical results on MNIST digits are provided, since it is almost standard in existing literatures. It would be better if the empirical run time comparison of FW and SD can be provided, so we can have a more intuitive understanding of the speed of the method. ------------------------------------------------------ The authors have added more empirical result. And promise to add more discussion on running time. I increased my score correspondingly.

Correctness: I checked the contents to the best of my ability. But it is still possible that I miss something in the proof.

Clarity: This paper flows well and is pleasant to read.

Relation to Prior Work: Yes. The discussion is concise and informative.

Reproducibility: Yes

Additional Feedback:

[Author Response · NeurIPS 2020]

We thank the reviewers for their careful consideration and constructive feedback. Please find our responses below.

**General response to all reviewers on empirical study of** SD. A common suggestion of all the reviewers for improving

the paper is to provide more empirical study of SD. We hence provide the results on the MNIST dataset. Specifically, we

consider 10 target distributions (one for each digit) and our goal is to compute their Sinkhorn barycenter (we randomly

select 100 images for each digit to construct the t.). We compare SD ($\gamma = 1e^{-4}$) with the free-support (FS) method,

i.e. Algorithm 2 in [Cuturi and Doucet, 2014]. We use the implementation from the PythonOT library, which does not

optimize over the weights (hence all particles have uniform weights like SD), nor it does local line searching. Note that

this is the same as the fix point iteration (section 5.3) in [Claici et al, 2018]. Both methods use 2500 particles. We do

not include the iterative Bregman projection (IBP) algorithm here since its implementation in PythonOT has numerical

issues for the small $\gamma$ case (it involves the division of $\exp(-1/\gamma)$). We will implement a numerical-stable version of

IBP and add the result in our revision. Note that IBP does not handle the debiasing term in the Sinkhorn barycenter

problem. [Chewi et al. 2020] is excluded here as it only applies to the barycenter problem of Gaussian distributions.

In figure (i), the first row is the results of SD and the second row is the results of FS. To interpret these plots, brighter

pixels mean more particles in a region. We can observe that the particles of SD are more concentrated on the digits

compared to the ones of FS. We present in figure (ii) the comparison of convergence rate between SD and FS (only for

digit 8–similar for others). Note that FS aims to solve the original Wasserstein barycenter problem without entropy

regularization. Consequently, evaluating FS using the Sinkhorn barycenter objective value may not be very precise.

(i) Top row is from SD and the bottom row is from FS

(ii) loss vs # iter

**To Reviewer #1.** Q1. About Remark 3.1. A1. A deeper understanding of the RKHS restriction on the push-forward

mapping is a very interesting direction and we are working on this right now. Q2. What if the variable measure $\alpha$ is

discrete? A2. In this case, the result of Theorem 4.1 still stands and SD converges to a stable point. However, there is no

guarantee for quality of such stable point: SD will *not* converge to the global optimal since $\alpha$ is not fully supported and

the assumption in Theorem 4.3 does not hold. Q3. More experiments. A3. Please see our general response above. Q4.

The limitation of discrete initialization $\alpha_0$. A4. This is an excellent point raised by the reviewer. In practice, we start

with a sufficient number of particles. Besides, we observe that increasing the number of particles reduces the Sinkhorn

divergence at convergence, which, however, has diminishing returns. Q5. The gradients correspond to the extension of

the potentials. A5. We will mention this and correct other typos in our revision. Thanks.

**To Reviewer #2.** Q1. Dependence on $\exp(1/\gamma)$. A1. Indeed, this problem is believed to be hard in the literature. The

term $\exp(1/\gamma)$ appears in bounding the derivatives of Sinkhorn potentials, which also appears in bounding the sample

complexity of the Sinkhorn divergence (see Theorem 2 and Lemma 3 of [Genevay et al. 2019]). The sample complexity

in [Genevay et al. 2019] can be improved if one manages to remove this factor. However, this would potentially violate

the lower bound on the sample complexity of the hard-to-compute Wasserstein distance, since it is the limit of the

Sinkhorn divergence at $\gamma \to 0$. We will elaborate on this in our revision. Q2. How does $\exp(1/\gamma)$ impact practice. A2.

Surprisingly, the empirical performance of SD does not suffer much from this factor: In our experiments, to produce

good visual results, we pick $\gamma = 10^{-4}$ and we still observe that SD quickly converges (even in the high dimensional

Gaussian barycenter task). We observe that the problem can be solved to high accuracy with different configurations of

$\gamma$. Besides, a larger $\gamma$ results in a more blurred barycenter. We will elaborate more on the impact of $\gamma$ in our revision.

Q3. Comparison with iterative Bregman projection. A3. Please see our general response above. Q4. logsumexp in SD.

A4. Yes. Q5. y-axis of Figures 1a and 1c. A5. Both y-axes are the Barycenter function values (the latter).

**To Reviewer #3.** Q1. Implicit exponential dependence on the problem dimension. A1. Indeed, as shown in Lemma 4.1

and Theorem 4.1 (see line 209 and 214), our results depend on $\exp(M_c/\gamma)$ where $M_c$ is the upper bound on ground

cost on the domain $\mathcal{X}$ which contains an implicit dependence on the problem dimension. We will elaborate on this in

our revision. We will also discuss that Sinkhorn divergence interpolates Wasserstein distance and MMD. Q2. cost of

[Genevay et al. 2016]. A2. Since the entropy regularized optimal transport problem is strongly convex, SGD converges

at the rate $\mathcal{O}(1/k)$ for such problem, where $k$ is the number of SGD steps. Besides, the per-iteration complexity is

$O(n^2)$ where $n$ is the support size of input measures. We will discuss about it in the revision. Q3. Comparison with

classical tools for computing barycenter. A3. Please see our general response above. Q4. Missing citation. A4. Thanks.

We will properly cite all the works mentioned by the reviewer in our revision. Q5. Regularization parameter. A5. We

set $\gamma = 10^{-4}$ in all of our experiments to produce results of good visual quality.

**To Reviewer #4.** Q1. Empirical result on MNIST. A1. Please see our general response above. Q2. Running time

comparison of FW and SD. A2. In our experiment, we directly use the implementation of FW from the original paper and

we observe that SD is much more efficient than FW. This is because each FW step requires to globally solve a nonconvex

subproblem via grid search as discussed in lines 258-263 of our paper. We will highlight this in our revision.

[Meta-Review · NeurIPS 2020]

This paper proposes a new method to compute the (Sinkhorn) of barycenter of several probability measures. In practice, the method scales well computationally and in high-dimension and the authors provide some theoretical support. Reviewers agree that this paper is strong with only minor weaknesses (such as the exponential dependency in 1/gamma; which in related work is often suboptimal). I thus recommend accept (poster).